# SEPARATION-UTILITY PARETO FRONTIER: AN INFORMATION-THEORETIC CHARACTERIZATION

## ABSTRACT

We study the Pareto frontier (optimal trade-off) between utility and separation, a fairness criterion requiring predictive independence from sensitive attributes conditional on the true outcome. Through an information-theoretic lens, we prove a characterization of the utility-separation Pareto frontier, establish its concavity, and thereby prove the increasing marginal cost of separation in terms of utility. In addition, we characterize the conditions under which this trade-off becomes strict, providing a guide for trade-off selection in practice. Based on the theoretical characterization, we develop an empirical regularizer based on conditional mutual information (CMI) between predictions and sensitive attributes given the true outcome. The CMI regularizer is compatible with any deep model trained via gradient-based optimization and serves as a scalar monitor of residual separation violations, offering tractable guarantees during training. Finally, numerical experiments support our theoretical findings: across COMPAS, UCI Adult, UCI Bank, and CelebA, the proposed method substantially reduces separation violations while matching or exceeding the utility of established baseline methods. This study thus offers a provable, stable, and flexible approach to enforcing separation in deep learning.

## 1 INTRODUCTION

Automated decision systems are increasingly deployed in high-stakes domains such as finance, criminal justice, hiring, and healthcare, making verifiable fairness constraints an essential component of reliable machine learning Barocas & Selbst (2016). A central challenge is that fairness criteria often conflict with model utility Chouldechova (2017); Kleinberg et al. (2017), creating a critical need for a framework that provides practitioners with a provably optimal and practically implementable trade-off.

In this work, we focus on **separation** (or equalized odds in binary classification), which requires conditional independence between the model output $\widehat{Y}$ and a sensitive attribute $Z$ given the true label $Y$ Hardt et al. (2016):

$$\widehat{Y} \perp Z \mid Y.$$

Any dependence of $\widehat{Y}$ on $Z$ must be justified by the overlapping information between $Z$ and the target $Y$. Separation is particularly relevant when base rates differ across groups, as it rules out unjustified group-dependent errors while remaining compatible with perfect prediction.[*]

**Information-plane view and the Pareto frontier.** We take an information-theoretic perspective on the separation-utility trade-off by quantifying (i) separation violation $v$ by the conditional mutual information, and (ii) predictive utility $u$ by mutual information:

$$v(\widehat{Y}) := I(\widehat{Y}; Z \mid Y), \qquad u(\widehat{Y}) := I(\widehat{Y}; Y).$$

Here, $v$ characterizes separation at $v(\widehat{Y}) = 0$, whereas $u$ is tightly related to prediction accuracy upper bounds (via Fano's inequality) and provides a lower bound on common loss functions such as Cross-Entropy and MSE (via Rate-Distortion Theory) Cover & Thomas (2006).

---

[*]We adopt separation as the fairness definition throughout. Comparing separation to alternative notions (e.g., independence, calibration) is beyond the scope of this paper.

This formulation yields a model-agnostic feasibility region on the information plane:

$$\mathcal{R} := \big\{ (v(\widehat{Y}), u(\widehat{Y})) : \ \widehat{Y} \text{ feasible from a predictor} \big\}.$$

Crucially, this theoretical characterization is general: It holds for arbitrary distributions of $(\widehat{Y}, Y, Z)$, regardless of whether they are continuous, discrete, or mixed. Our first goal is to characterize the shape of the Pareto frontier of $\mathcal{R}$, thereby establishing the "price of separation" in terms of utility.

**From theory to practice: An empirical regularizer.** While the theoretical frontier is general, practical optimization requires estimating CMI. In continuous settings, this often necessitates complex auxiliary density models or variational bounds. However, we observe that for the widespread case of discrete fairness tasks, ranging from standard classification to high-cardinality discrete output spaces, such complexity is unnecessary. To bridge the gap between theory and practice, we propose a simple in-processing regularizer that directly penalizes the empirical plug-in estimator of CMI. Unlike learning-based proxies, this sample statistics-based approach is stable, transparent, and recovers the theoretical Pareto frontier more robustly than the more sophisticated learning-based baselines, as we later show empirically in Section 4.

## 1.1 RELATED WORKS

Existing approaches for enforcing separation generally fall into three paradigms, each with distinct limitations regarding stability and scalability.

**Rate-constraints and reductions.** Reductions convert fairness constraints into cost-sensitive classification problems Agarwal et al. (2018); Cotter et al. (2019), while convex frameworks employ covariance-based surrogates Celis et al. (2019); Zafar et al. (2017). Although offering provable guarantees, they typically match only lower-order moments (mostly due to their focus on binary classification for which lower-order moments are sufficient) rather than the full distribution, and constraint complexity scales linearly with the product of target and sensitive feature cardinality, limiting scalability.

**Adversarial and information-theoretic proxies.** Adversarial debiasing uses min-max games to infer $Z$ from predictions (Zhang et al., 2018; Madras et al., 2018). Similarly, information-theoretic approaches penalize Conditional Mutual Information (CMI) via learning-based approximations, such as auxiliary density models (Steinberg et al., 2020) or variational bounds (Roh et al., 2020; Kang et al., 2023). While popular, these methods suffer from lack of theoretical guarantee, optimization instability (e.g., vanishing gradients, non-convexity), and reliance on the expressivity of the adversarial or auxiliary model, often introducing estimation errors (Song et al., 2020).

**Reweighting and robustness.** This line of works enforce fairness through external interventions on the data. *Distributionally Robust Optimization (DRO)* Sagawa* et al. (2020) targets the *worst-performing* subgroup, dynamically prioritizing high-loss regions. In contrast, *resampling* techniques Kamiran & Calders (2012) address group *imbalance* via static re-weighting, while *data transformation* methods Romano et al. (2020) introduce randomized variables or modify features to break statistical dependencies. Although model-agnostic, these methods can be unstable or degrade utility due to their aggressive manipulation of the training distribution.

## 1.2 CONTRIBUTIONS

Despite recent progress, three critical gaps remain. First, while fairness-utility trade-offs are well-documented (Pleiss et al., 2017; Chouldechova, 2017; Corbett-Davies et al., 2017; Menon & Williamson, 2018), the field lacks a general model-agnostic characterization of Pareto frontier beyond binary classification setting, particularly on its existence, shape, and finite-sample feasibility. Second, applicability is limited: most methods are tailored to binary tasks (Hardt et al., 2016; Agarwal et al., 2018), becoming intractable in modern high-cardinality settings (e.g., LLMs) where constraint complexity explodes (Gallegos et al., 2024; Chu et al., 2024; Blodgett et al., 2020). Finally, training lacks transparent guarantees: standard proxies (adversaries, variational bounds) neither directly certify separation (Zhang et al., 2018; Madras et al., 2018; Roh et al., 2020) nor ensure optimization stability (Agarwal et al., 2018; Duchi et al., 2018; Sagawa* et al., 2020).

We address these challenges through a cohesive narrative of practice-inspired theory, theory-guided method, and empirical verification:

- **Foundational characterization of the Pareto frontier.** We provide a general, model-agnostic theoretical foundation by characterizing the separation-utility region on an information-theoretic plane. We prove that the optimal randomized frontier equals the concave closure of the deterministic frontier (Theorem 2.2). This result holds for continuous, discrete, or mixed variables, clarifying which trade-offs are information-theoretically feasible independent of any optimization surrogate. This provides a theoretical common ground for existing and future work that targets separation via information-theoretic regularization, validating the objective regardless of how it is estimated.
- **CMI as a principled separation quantification.** We show that $I(\widehat{Y}; Z \mid Y)$ provides exact characterization of separation and an uniform upper bound controlling the conditional statistical dependence of *all* bounded functionals (Theorem 2.5). This positions CMI as a principled scalar quantification for separation violation.
- **Direct empirical CMI regularization.** Bridging the gap between optimization and theoretical guarantees, we demonstrate that complex learning-based proxies are often unnecessary. We show that for discrete separation tasks, the target information-theoretic quantity admits a direct sample estimator derived from empirical frequencies. This enables transparent statistical guarantees under standard i.i.d. assumptions, bypassing the inherent instability of min-max optimization and the estimation errors of variational bounds.
- **Empirical verification.** We evaluate our approach on standard benchmarks, validating that our theoretical findings translate into practice. This simple statistics-based regularizer yields smooth, stable Pareto frontiers that often outperform computationally complex reduction-based, adversarial, or learning-based baselines in terms of both robustness and the achieved trade-off. [†]

## 2 METHODS: REGULARIZATION WITH THEORETICAL GUARANTEES

We present a principled regularization framework for enforcing *separation* during training. To maintain consistency and clarity, we first introduce the setting and notation, then develop the theoretical results in the following order: **Section 2.1**: The characterization of the separation-utility Pareto frontier; **Section 2.2**: The justification for Conditional Mutual Information (CMI) as a rigorous measure of separation violation; **Section 2.3**: The conditions under which a trade-off is unavoidable.

**Setting and notation.** Let $(\mathcal{X}, \mathcal{B}_{\mathcal{X}})$, $(\mathcal{Y}, \mathcal{B}_{\mathcal{Y}})$, and $(\mathcal{Z}, \mathcal{B}_{\mathcal{Z}})$ be Polish spaces equipped with Borel $\sigma$-algebras. The random variables $(X, Y, Z)$ are jointly distributed according to $P_{XYZ}$ and admit regular conditional laws. Here, $X$, $Y$, and $Z$ denote the input, target, and sensitive variables.

A deterministic predictor is a measurable map $f : \mathcal{X} \times \mathcal{Z} \to \mathcal{Y}$, producing the output $\widehat{Y} = f(X, Z)$. A randomized predictor is defined by an external independent noise variable $N$ (with $N \perp (X, Y, Z)$) and a measurable map $f$, producing the augmented variable $\widetilde{U} = (N, \widehat{Y})$ where $\widehat{Y} = f(X, Z, N)$.

To simply notation, we use a generic prediction $U$ to represent either $\widehat{Y}$ or $\widetilde{U}$, with the generic space $\mathcal{U}$ representing either the outcome space $\mathcal{Y}$ or the product space $\mathcal{N} \otimes \mathcal{Y}$. We define *predictive utility* ($u$) and *separation violation* ($v$) for the generic predictor variable $U$ as:

$$u(U) := I(U; Y), \qquad v(U) := I(U; Z \mid Y).$$

Note that, for a randomized predictor $\widetilde{U}$, $I(\widetilde{U}; Y) = I(\widehat{Y}; Y)$ due to the independence of the randomness source $N$. We maximize the mutual information because it minimizes the expected log-loss and bounds the minimum error probability via Fano's inequality. Separation (equalized odds) is strictly satisfied when $v(U) = 0$, implying conditional independence. For theoretical bounds, we assume finite entropies $H(Y)$, $H(Z \mid Y)$, and mutual information $I((X, Z); Y)$.

### 2.1 THE SEPARATION-UTILITY PARETO FRONTIER

We begin by characterizing the set of feasible (violation, utility) pairs. This characterizes the theoretical limit of what is achievable. Define the deterministic attainable set $\mathcal{S}_{\text{det}}$ and the randomized attainable set $\mathcal{S}_{\text{rand}}$ as follows:

- Let $u_f := I(f(X, Z); Y)$ and $v_f := I(f(X, Z); Z \mid Y)$. The deterministic set is:

$$\mathcal{S}_{\text{det}} := \left\{ (v_f, u_f) : f \text{ is measurable} \right\} \subset \mathbb{R}_{\geq 0}^2.$$

---

[†]Code is available upon acceptance.

- Let $\widetilde{U} := (N, f_N(X, Z))$ be a randomized predictor where $N \perp (X, Y, Z)$. The randomized set is:

$$\mathcal{S}_{\mathrm{rand}} := \big\{ (I(\widetilde{U}; Z \mid Y),\, I(\widetilde{U}; Y)) : \widetilde{U} \text{ is randomized} \big\}.$$

We define the corresponding Pareto frontiers $U_{\mathrm{det}}^{\star}(v)$ and $U_{\mathrm{rand}}^{\star}(v)$ as the maximum utility attainable for a given violation constraint $v$:

$$U_{\mathrm{det}}^{\star}(v) := \sup_{\substack{f \text{ measurable} \\ I(f(X,Z);Z|Y) \leq v}} I\big(f(X, Z); Y\big), \qquad U_{\mathrm{rand}}^{\star}(v) := \sup_{\substack{\widetilde{U} \text{ randomized} \\ I(\widetilde{U};Z|Y) \leq v}} I(\widetilde{U}; Y).$$

First, it is clear that the deterministic frontier is non-decreasing.

**Proposition 2.1** (Deterministic frontier is non-decreasing). *For any $0 \leq v_1 \leq v_2 \leq H(Z \mid Y)$, we have*

$$U_{\mathrm{det}}^{\star}(v_1) \leq U_{\mathrm{det}}^{\star}(v_2).$$

See proof in Appendix B.5. This implies that decreasing separation violation can potentially harm utility, but the trade-off is not guaranteed to be convex or smooth for deterministic predictors.

Now, we prove that the randomized frontier is concave:

**Theorem 2.2** (Randomized frontier equals the concave closure). *The set $\mathcal{S}_{\mathrm{rand}}$ contains the convex hull of $\mathcal{S}_{\mathrm{det}}$ and is contained in its closure:*

$$\mathrm{conv}\big(\mathcal{S}_{\mathrm{det}}\big) \subset \mathcal{S}_{\mathrm{rand}} \subset \overline{\mathrm{conv}}\big(\mathcal{S}_{\mathrm{det}}\big). \tag{1}$$

*Consequently, the randomized upper frontier is the concave closure of the deterministic frontier:*

$$U_{\mathcal{S}_{\mathrm{rand}}}(v) = U_{\overline{\mathrm{conv}}(\mathcal{S}_{\mathrm{det}})}(v) \quad \text{for all } v \geq 0. \tag{2}$$

See a proof in Appendix B.6. The result shows two key messages: (1) The randomized Pareto frontier is concave, meaning the *marginal cost* of reducing separation is non-decreasing in terms of utility loss. That is, every unit of further reduction in separation violation requires a progressively larger sacrifice in terms of utility units. (2) Any point $(v, u)$ on the optimal randomized frontier can be achieved by mixing at most two deterministic predictors (e.g., via Bernoulli randomization).

### 2.2 Information-Theoretic Quantification of Separation Violation

Having characterized the frontier on the information plane, we now justify why conditional mutual information (CMI) is the correct quantification for separation violation ($v$).

To start, we show that zero CMI characterizes the separation definition:

**Proposition 2.3** (CMI characterizes separation).

$$I(U; Z \mid Y) = 0 \quad \Longleftrightarrow \quad U \perp Z \mid Y.$$

See a proof in Appendix B.4. Beyond simple characterization, CMI provides a strong statistical guarantee: it controls the conditional dependence for *all* bounded test functions (e.g., auditors or queries). the normalized covariance for bounded maps $h : \mathcal{U} \to \mathbb{R}^d$ and $g : \mathcal{Z} \to \mathbb{R}^d$:

$$\rho(h, g) := \frac{\big| \langle h(U) - \mathbb{E}(h(U)), g(Z) - \mathbb{E}(g(Z)) \rangle_{L^2} \big|}{\|h(U)\|_{L^\infty} \|g(Z)\|_{L^\infty}}. \tag{3}$$

**Lemma 2.4** (Mutual information controls dependence). *For any bounded measurable $h, g$ with zero mean,*

$$\sup_{h,g} \rho(h, g) \leq \sqrt{2\, I(U; Z)}. \tag{4}$$

See a proof in Appendix B.1. Extending this to the conditional setting yields our main guarantee:

**Theorem 2.5** (CMI controls average conditional dependence). *Let $h, g$ be bounded measurable functions such that $\mathbb{E}[h(U)|Y = y] = 0$ and $\mathbb{E}[g(Z)|Y = y] = 0$ almost surely. Then:*

$$\sup_{h,g} \frac{\mathbb{E}_y\big[ \big| \langle h(U), g(Z) \rangle_{\mathcal{L}(\cdot|y)} \big| \big]}{\|h\|_{L^\infty} \|g\|_{L^\infty}} \leq \sqrt{2\, I(U; Z \mid Y)}. \tag{5}$$

This confirms that minimizing CMI uniformly suppresses any conditional correlation an adversary could detect between the prediction and the sensitive attribute.

## 2.3 When Is a Trade-off Necessary?

Now, we examine the conditions under which increasing utility *necessarily* forces an increase in separation violation. Let $U$ denote the generic learning outcome. Recall our primary coordinates: utility $u = I(U; Y)$ and separation violation $v = I(U; Z \mid Y)$.

We first establish that the sum of utility and violation is constrained by a universal information budget:

**Lemma 2.6** (Budget identity and universal bounds)**.** *For a generic predictor $U$,*

$$u + v \;=\; I(U; (Y, Z)) \;\leq\; I((X, Z); (Y, Z)) \;=\; I((X, Z); Y) + H(Z \mid Y).$$

*Hence, $0 \leq u \leq I((X, Z); Y)$ and $0 \leq v \leq H(Z \mid Y)$. If the output space $\mathcal{Y}$ is finite, then $u \leq \log |\mathcal{Y}|$.*

See a proof in Appendix B.7. This budget identity reveals that, in full generality, there is no necessary strict trade-off. There are degenerate cases where perfect utility and perfect fairness coexist. For instance, if $Y = (X, Z)$, then $I((X, Z); Y) = H(Y)$ while $H(Z \mid Y) = 0$. That is, the information budget is entirely allocated to utility without incurring any separation penalty.

To rule out such degeneracy and analyze the realistic setting where conflict often arises, we impose the following non-degeneracy condition for the remainder of the analysis:

$$X \perp Z \mid Y, \qquad \text{and} \qquad Y \not\perp Z \mid X. \tag{6}$$

- The condition $X \perp Z \mid Y$ implies that $Z$ carries no information about $X$ that is not already mediated by $Y$. While this may not hold for raw data, we note that one can always deform $(X, Z)$ into $(X', Z)$ via a conditional Wasserstein barycenter characterization Xu & Strohmer (2023) to achieve $I((X, Z); Y) = I((X', Z); Y)$ while satisfying $X' \perp Z \mid Y$ Xu & Strohmer (2025). We adopt this assumption to simplify the analysis by decoupling the information overlap between $X$ and $Z$ beyond $Y$.
- The condition $Y \not\perp Z \mid X$ rules out the degenerate cases where $Z$ is redundant given $X$. It ensures that $Z$ contributes unique predictive power for $Y$ beyond what is contained in $X$.

Under assumptions equation 6, any predictor that achieves perfect separation ($v = 0$) cannot "use" $Z$ in any way that alters its conditional law given $Y$. The following lemma formalizes this intuition: a perfectly fair predictor must essentially ignore $Z$.

Let us define the optimal utility achievable using only $X$ versus using both $X$ and $Z$. Let $U_X \in \{f(X), (N, f(X, N))\}$ denote the generic prediction that only explicitly uses $X$ or $X$ and $N$, but not $Z$. Define

$$u_X^\star := \sup_{U_X} I(U_X; Y), \qquad u_{XZ}^\star := \sup_U I(U; Y).$$

**Lemma 2.7** (Conditional law matching)**.** *Let $(\mathcal{X}, \mathcal{Y}, \mathcal{Z}, \mathcal{U})$ be Polish spaces, and let $(X, Y, Z)$ be a random vector satisfying $X \perp Z \mid Y$. Let $U \in \{f(X, Z), (N, f(X, Z, N))\}$ be any generic prediction that satisfies perfect separation, i.e., $I(U; Z \mid Y) = 0$. Then, there exists a measurable set $\mathcal{Z}_0 \subseteq \mathcal{Z}$ satisfying $\mathbb{P}(Z \in \mathcal{Z}_0) = 1$ and, for every $z_0 \in \mathcal{Z}_0$, $\mathcal{L}\big(f(X, z_0) \mid Y = y\big) = \mathcal{L}\big(U \mid Y = y\big)$ for $P_Y$-almost all $y$. Consequently, the joint laws $\mathcal{L}\big(f(X, z_0), Y\big)$ and $\mathcal{L}\big(U, Y\big)$ are identical $P_Z$-almost surely, implying*

$$I\big(f(X, z_0); Y\big) = I(U; Y) \qquad P_Z\text{-almost surely.}$$

See proof in Appendix B.8. This leads to our main necessity result: in the ideally decoupled setting where $X \perp Z \mid Y$, a trade-off is unavoidable whenever $Z$ holds unique predictive power for $Y$.

**Theorem 2.8** (Necessary trade-off beyond $u_X^\star$)**.** *Let $(\mathcal{X}, \mathcal{Y}, \mathcal{Z}, \mathcal{U})$ be Polish spaces and let $(X, Y, Z)$ be a random vector. It follows:*

- ***Fairness limit:*** *If $X \perp Z \mid Y$, then the maximum utility compatible with perfect separation is exactly the utility of the best $X$-only predictor:*

$$\sup_{U_X} \Big\{ I(U_X; Y) : \ I(U_X; Z \mid Y) = 0 \Big\} = u_X^\star.$$

- **Strict trade-off:** *In addition, if $I(Z;Y \mid X) > 0$, then there is a necessary trade-off between increasing $u$ and decreasing $v$. Specifically, for any generic prediction $\widetilde{U}$ to achieve utility $u > u_X^\star$, one must incur strictly positive violation:*

$$v = I(U; Z \mid Y) > 0.$$

See proof in Appendix B.9. Without equation 6, the conclusion can fail. For instance, if $Y = Z$, then $U = Y$ gives $v = 0$ while $u$ may exceed $u_X^\star$. Assumption equation 6 excludes such $Z$-mediated shortcuts: any $v=0$ predictor must (a.s.) be a function of $X$ alone.

The results above provide practical guidance for the trade-off between utility and separation. In practice, one often first obtains $u_{XZ}^*$, which serves as the right-most endpoint of the Pareto frontier on the information plane $\{(u, v)\}$. If one then obtains $u_X^*$, the linear interpolation between $u_{XZ}^*$ and $u_X^*$ provides an *upper-bound* baseline for the rest of the frontier, given that the frontier is concave as shown in Theorem 2.2. In this context, the difference $|u_{XZ}^* - u_X^*|$ serves as a linear estimate of the price of separation on the given data set at the very beginning of the Pareto frontier, representing how much unique information $Z$ contributes to $Y$ beyond $X$. Due to the concavity of the frontier, this is the cheapest and most conservative estimate of the "price of fairness" for the given task on the given dataset.

## 3 TRAINING WITH DIRECT CMI REGULARIZATION

Guided by the theory in Section 2, we train with a direct conditional mutual information (CMI) penalty:

$$\mathcal{L}_{\text{total}} \;=\; (1-\gamma)\,\frac{\mathcal{L}_{\text{task}}}{\|\nabla \mathcal{L}_{\text{task}}\|} \;+\; \gamma\,\frac{\widehat{I}_{\text{CMI}}}{\|\nabla \widehat{I}_{\text{CMI}}\|}, \tag{7}$$

where $L_{\text{task}}$ is the standard prediction loss, and $\widehat{I}_{\text{CMI}}$ is a differentiable mini-batch estimator of $I(U; Z \mid Y)$. The gradient–norm normalization stabilizes optimization by balancing the update magnitudes of the utility and fairness terms.

All implementation details, including the sample estimator of CMI, full pseudocode, and finite, sample bias/concentration guarantees for the estimator, are deferred to Appendix C.

## 4 NUMERICAL EXPERIMENT

This section provides empirical evidence for the separation-utility theory in Section 2 and evaluates the proposed *Normalized CMI* regularization in Section 3. Across four benchmarks (Adult, COMPAS, Bank, CelebA), the experiments are designed to test four contributions of the paper:

1. **Frontier geometry (theory leads to observable prediction).** We empirically recover the theoretically proved concave separation-utility frontier and show that allowing *randomized policies* yields a smoother and typically higher envelope than *deterministic* (thresholded) policies, consistent with the randomization/concavification result in Theorem 2.2.
2. **Stable travel on the frontier (algorithmic contribution).** Tuning the trade-off parameter yields a monotonic Pareto frontier estimated by CMI and the resulting curve exhibits smaller fold variance, especially in the strict regime near zero separation violation where baselines often show backtracking, range collapse, or variance blow-up. This empirically supports the advantages in guaranteed stability of statistics-based estimation shown in Proposition C.1, when compared with the learning-based proxies.
3. **Operational transfer (information plane generalizes to deployment metrics).** Improvements in the information plane, lower $I(U; Z \mid Y)$ at comparable $I(U; Y)$, transfer to improved Equalized Odds (EO) gap at comparable Accuracy/AUROC on test data folds, demonstrating that CMI is a generalizable separation quantification for separation violations beyond itself. This provides empirical evidence to advantage of the uniform (in arbitrary downstream function covariance) upper bound guarantee shown in Theorem 2.5.
4. **Representation-level separation vs. post-hoc thresholding.** On CelebA, the randomized evaluation (posterior-level) cleanly separates methods: CMI reduces sensitive dependence in the posterior itself, while several baselines appear competitive only after deterministic thresholding, illustrating that post-hoc decision repair can mask representation-level violations.

### 4.1 Setup, Metrics, and Baselines

**Datasets.** We evaluate on four standard fairness benchmarks: Adult, COMPAS, Bank, and CelebA, which cover tabular and image modalities with varying sample sizes and imbalances. For all datasets, $X$ denotes input features, $Y$ the target variable, and $Z$ the sensitive attribute. See detailed description of the datasets in Table 1, Appendix D.1.

**Metrics.** We report Utility via Mutual Information $I(U; Y)$ and Accuracy/AUROC respectively as the theoretical and practical/deployment metrics. We report Separation Violation via Conditional Mutual Information $I(U; Z \mid Y)$ and Equalized Odds Gap EO Gap respectively as the theoretical and practical metrics. The purpose of the CMI-MI plot is to verify the theoretically proved concave Pareto frontier, whereas the Accuracy-EO Gap plot aims to demonstrate the principled information theoretic Pareto frontier generalizes and transfers well to practical metrics in deployment. See Appendix D.2 for more details and formal definitions of the metrics.

**Baselines.** We compare CMI against seven state-of-the-art methods: (1) Constraint-based: *EG Reductions (denoted by ExpGrad)* Agarwal et al. (2018); (2) Adversarial: *Adversarial Debiasing (denoted by Adversarial)* Madras et al. (2018), *Fair Dummy (denoted by FairDummy)* Romano et al. (2020); (3) Info-Theoretic Proxies: *FR-Train (denoted by FR-Train)* Roh et al. (2020), *InfoFair (denoted by InfoFair)* Kang et al. (2023); (4) Robustness: *FairDRO (denoted by FairDRO)* Park et al. (2025). All methods use a unified MLP backbone. We report results for both Randomized Policies (raw posterior, evaluating representation) and Deterministic Policies (thresholded, evaluating decision rule). See Appendix D.3 for more detailed explanation of the baselines.

**Fair comparison protocol.** To isolate the efficacy of the regularization objective, we standardize the training pipeline: (1) unified backbone: all methods use the same MLP architecture (two hidden layers). (2) randomized vs. deterministic evaluation: to compare ranking quality versus thresholding effects. *Randomized Policy*: The raw posterior probability $P(Y = 1 \mid X = x)$. This evaluates the inherent separation of the learned representation. *Deterministic Policy*: Hard predictions via a threshold $t^*$ tuned on validation data to maximize Accuracy $- \lambda \cdot$ EO gap to simulate real-world decision practice. This evaluates the final decision rule. (3) Cross-Validation: We report mean $\pm$ 1 standard deviation over 5 stratified folds.

### 4.2 Results & Analysis

We analyze the Pareto frontiers generated by tuning the fairness trade-off parameter for each method.

#### 4.2.1 Bank Dataset: Frontier Shape & Operational Transfer

Figure 1 reveals that on the Bank dataset, the information plane trade-off is concentrated in an extremely small-violation region. In the randomized information plane (top-left), most methods exhibit a very steep local slope near the origin: moving from (effectively) zero violation to a tiny positive violation yields a rapid jump from a near-trivial predictor to a high-utility regime where $I(U; Y)$ saturates.

This suggests a nuanced interpretation of the "price of separation" for this learning task on Bank: the marginal cost of separation varies dramatically. Therefore, it is important for practitioners to carefully select an ideal balance point on the frontier for the specific learning task and dataset. In this particular case, strict separation can be too expansive, while permitting arbitrarily small violations recovers near-saturated utility.

- **Stability.** Regarding optimization stability, while several methods (e.g., FairDummy, Adversarial) show comparable variance in specific regimes, CMI yields a consistently smooth and monotonic estimation of the Pareto frontier that reliably traces the dominant envelope, especially in the low-violation region. Crucially, it avoids the significant fold-level variance and occasional collapse observed in learning-based baselines, confirming that the direct sample-based CMI estimator provides a robust signal even in this sensitive, low-violation regime.
- **Operational transfer.** The strongest empirical evidence on Bank appears in the operational transfer plots (bottom row). As the Equalized Odds (EO) gap is tightened toward zero, CMI maintains high Accuracy/AUROC while remaining close to the strict-separation end of the curve. In contrast, learning-based baselines often exhibit larger variance or severe utility degradation in

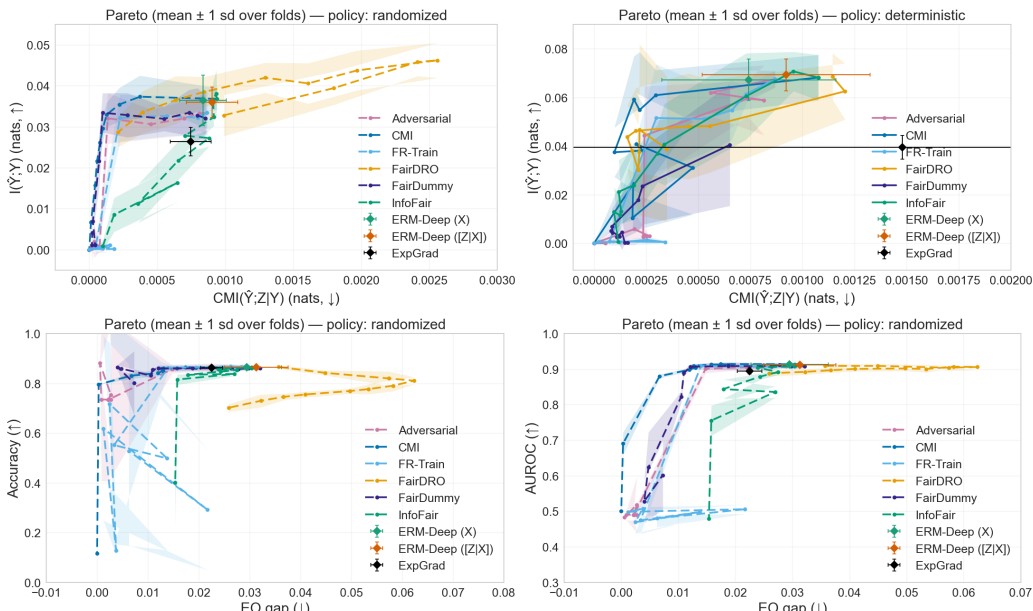

Figure 1: **Bank Results: High Price of Fairness and Strict-Regime Stability. Top (Information Plane):** The frontier estimation reveals a steep "knee" near the origin, indicating a *dramatically varying marginal cost of separation*. Normalized CMI provides a smooth, stable estimation of this dominant envelope, avoiding the significant fold-level variance or collapse observed in the comparison baselines near the strict-separation boundary. **Bottom (Operational Transfer):** These advantages transfer to superior performance in deployment metrics on test fold. CMI maintains high Accuracy and AUROC specifically in the critical strict-fairness regime (low EO gap), validating theoretical generalization guarantees.

the small EO gap regime. This empirically validates our theoretical result in Theorem 2.5 (uniform covariance upper bound) and sample estimation concentration (see Proposition C.1 in Appendix C), together implying that penalizing the empirical CMI effectively generalizes to deployment metrics, particularly when strict fairness is required.

- **Randomized vs. deterministic evaluation.** Finally, consistent with Theorem 2.2, the randomized evaluation yields a cleaner Pareto frontier than deterministic thresholding. The latter introduces additional irregularity due to post-hoc threshold selection, obscuring the representation-level (model's intrinsic probability distribution or belief) trade-off. Thus, we interpret the randomized curves as the primary evidence for separation capacity, and deterministic curves as a deployment-oriented diagnostic.

### 4.2.2 CelebA Dataset: Scalability & Posterior Separation

Figure 2 uses CelebA to test the framework on high-dimensional image inputs. While the prediction target remains binary, the optimization challenge lies in regularizing the high-dimensional feature representation (model internal parameter representation, e.g. the penultimate layer). This benchmark isolates the method's scalability to deep representation learning, distinguishing methods that can effectively regularize complex embeddings from those that degrade due to the dimensionality of the latent space.

- **Scalability to high-dimensional embeddings.** In the information plane (top-left), CMI dominates the envelope with a smooth Pareto frontier. It drives $I(U; Z \mid Y)$ close to zero without the *range collapse* or *instability* observed in comparison baselines. This offers critical empirical evidence that, unlike min-max or adversarial objectives which often struggle to converge when the *embedding dimension* is high, the direct CMI estimator provides a stable, well-conditioned gradient signal for separating deep posterior representations.
- **The "masking" effect & operational transfer.** The operational plots (bottom row) confirm that these representation-level gains transfer to superior deployment metrics trade-offs (Accuracy vs. EO gap) on test data. More importantly, the discrepancy between the randomized (top-left) and deterministic (top-right) frontiers is most acute on this dataset. While thresholding compresses the apparent performance gaps (top-right), the randomized view reveals that baselines rely heavily on post-hoc correction to mask posterior dependence. CMI, on the other hand, achieves separation

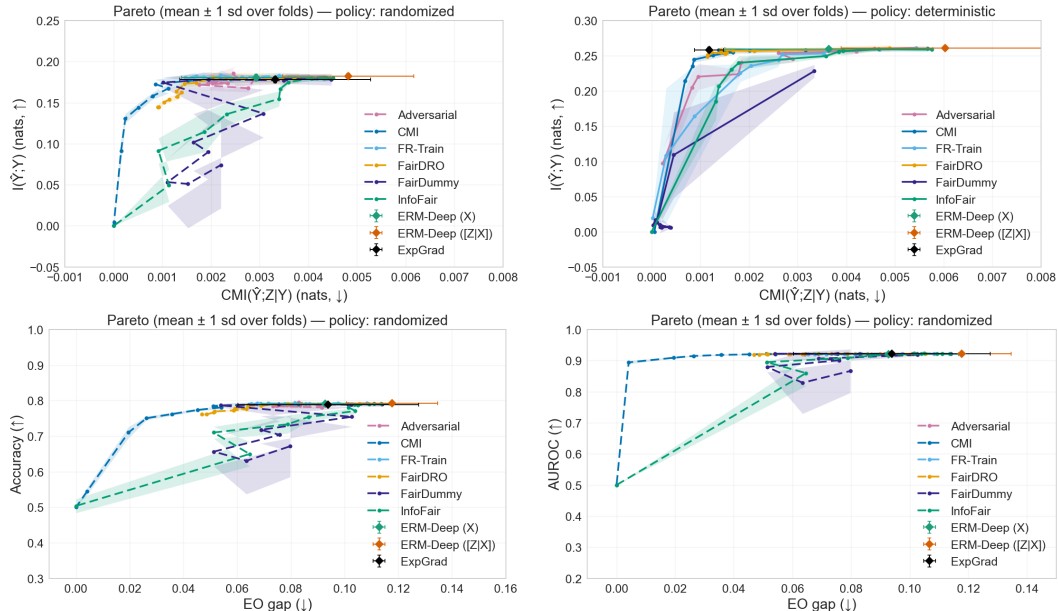

Figure 2: **CelebA Results: Scalability and Posterior Separation. Top (Information Plane):** In the randomized view (left), Normalized CMI dominates the envelope, avoiding the range collapse and instability seen in baselines on high-dimensional embeddings. The narrowing of separation in the deterministic view (right) confirms that post-hoc thresholding can "mask" underlying posterior dependence. **Bottom (Operational Transfer):** Operational metrics mirror the randomized frontier: CMI achieves superior Accuracy/AUROC specifically in the strict-fairness regime (small EO gap), confirming that better posterior separation translates to robust deployment performance.

in the posterior itself, validating its effectiveness in solving the harder problem of *intrinsic* bias mitigation in complex modalities.

### 4.2.3 CONSISTENCY ACROSS DOMAINS (ADULT & COMPAS)

We report full results for Adult and COMPAS in Appendix D.4 and D.5, respectively. These datasets corroborate the findings from Bank and CelebA, demonstrating the method's consistency across diverse tabular distributions.

**Frontier geometry & low marginal cost.** In contrast to the steep trade-off (in the small-violation region) observed on Bank, both Adult and COMPAS exhibit a more stable marginal cost of separation. CMI traces a smooth, concave frontier toward exact separation ($v = 0$) without the dramatic utility loss seen in the Bank. This confirms that the CMI objective adaptively identifies the optimal geometry of the problem, whether the trade-off is steep (Bank) or flat (Adult), without requiring manual tuning of the estimator itself.

**Robust transfer.** On both datasets, the "transfer" from information plane to deployment metrics holds: predictors that minimize the information-theoretic bound $I(U; Z \mid Y)$ reliably achieve the best EO-Accuracy trade-offs on held-out data. Specifically on COMPAS, which is typically prone to optimization noise, CMI avoids the non-monotonic frontiers and collapse observed in several learning-based baselines.

## 5 CONCLUSION

This work bridges the gap between fundamental fairness limits and practical optimization by providing a model-agnostic characterization of the separation-utility Pareto frontier. We proved the concavity of this frontier (Theorem 2.2) and demonstrated that complex learned proxies are often unnecessary for classification. Instead, a direct empirical estimator of Conditional Mutual Information yields smoother frontiers, lower variance, and intrinsically fairer representations than more sophisticated baselines.[‡]

---

[‡]During the preparation of this work, the authors used large language model ChatGPT by OpenAI to refine the language and enhance readability. After using this tool or service, the authors reviewed and edited the content as needed and take full responsibility for the content of the publication.

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

CONTENTS

# A APPENDIX TO SECTION 1

## A.1 RELATED MACHINE LEARNING FAIRNESS DEFINITIONS

To start, we provide a list of the most important machine learning fairness definitions and discuss their application to LLMs regarding fairness, safety, and alignment.

**1. Predictive Parity (a.k.a. Independence)** Predictive parity is motivated by the idea that a model is fair if it provides the same prediction outcome, independent of its knowledge of sensitive information, such as gender or race in the typical fairness settings. Mathematically, a predictor $\hat{Y} := f(X)$ satisfies demographic parity Dwork et al. (2012) if

$$\hat{Y} \perp Z.$$

Equivalently, for any measurable event $A_{\mathcal{Y}} \in \mathcal{B}_{\mathcal{Y}}$ and any $z, z' \in \mathcal{Z}$, we have

$$P(\hat{Y} \in A_{\mathcal{Y}}|Z = z) = P(\hat{Y} \in A_{\mathcal{Y}}|Z = z'). \tag{8}$$

That is, statistical parity requires that the prediction distribution does not depend on the sensitive information $Z$. In the binary classification case, we have

$$P(\hat{Y} = 1 \mid Z = z) = P(\hat{Y} = 1), \forall z \in \mathcal{Z}.$$

This is potentially useful in LLMs in the case where one holds the belief that the prediction distribution of tokens or perplexity should be independent of the sensitive information, such as gender or race.

**2. Predictive Error Parity (a.k.a. Separation)** Predictive error parity is motivated by the idea that a fair model can make mistakes, but it should make the same mistake across all the sensitive groups or independent of the sensitive information. Mathematically, predictor satisfies separation if

$$\hat{Y} \perp Z|Y.$$

Equivalently, for any measurable event $A_{\mathcal{Y}} \in \mathcal{B}_{\mathcal{Y}}, y \in \mathcal{Y}$, and any $z, z' \in \mathcal{Z}$, we have

$$P(\hat{Y} \in A_{\mathcal{Y}}|Y = y, Z = z) = P(\hat{Y} \in A_{\mathcal{Y}}|Y = y, Z = z'). \tag{9}$$

In the binary classification case, the above reduces to equalized odds Hardt et al. (2016):

$$P(\hat{Y} = 1|Y = y, Z = z) = P(\hat{Y} = 1|Y = y)$$

for each $y \in \{0, 1\}$ and any $z \in \mathcal{Z}$.

There are several relaxations or alternatives of the conditional statistical parity:

- (1) Equal Opportunity requires only true positive rates to match

$$P(\hat{Y} = 1|Y = 1, Z = z) = P(\hat{Y} = 1|Y = 1).$$

- (2) Predictive Equality requires

$$P(\hat{Y} = 1|Y = 0, Z = z) = P(\hat{Y} = 1|Y = 0).$$

- (3) Performance Parity requries

$$\mathbb{E}\big[\ell(\hat{Y}, Y) \mid Z = z\big] = \mathbb{E}\big[\ell(\hat{Y}, Y)\big], \quad \forall z \in \mathcal{Z}.$$

- (4) Subgroup Fairness requires uniform control of error rates over a family $\mathcal{C}$ of subgroups $S$:

$$\max_{S \in \mathcal{C}} |P(\hat{Y} = 1 \mid Y = y, X \in S) - P(\hat{Y} = 1 \mid Y = y)| \le \epsilon, \forall y \in \mathcal{Y}.$$

This line of definitions is useful in LLMs in the case where one wants to avoid

**3. Predictive Truth Parity (a.k.a. Sufficiency)** The definition is motivated by the idea that the group that shares the same fair prediction should also share the same ground truth regardless of their sensitive information. Mathematically, a predictor satisfies predictive parity if

$$Y \perp Z|\hat{Y}.$$

In the binary classification case, it reduces to

$$P(Y = 1|\hat{Y} = y, Z = z) = P(Y = 1|\hat{Y} = y) \quad \forall z \in \mathcal{Z}, \quad \forall y \in \mathcal{Y}. \tag{10}$$

A definition that shares a similar idea is Multi-Calibration which requires the following: For a collection $\mathcal{C}$ of subgroups and discretized scores $v$, we have

$$\left|\mathbb{E}\big[Y \mid X \in S,\ \hat{p}(X) = y\big] - y\right| \le \epsilon, \quad \forall S \in \mathcal{C}, \quad \forall y \in \mathcal{Y}.$$

This line of definition can be useful in LLMs when one hopes to align the prediction to the known group-truth or knowledge.

## 4. OTHER FAIRNESS DEFINITIONS

Here, we list some important fairness definitions that cannot be categorized into the above three perspectives. We group them here because the definitions are very particular to the task of classification or rely heavily on the underlying metric of the spaces and, hence, they become difficult to be applicable to the LLMs.

- **Treatment Equality:** $\frac{P(\hat{Y}=0|Y=1,Z=z)}{P(\hat{Y}=1|Y=0,Z=z)} = $ constant across $z \in \mathcal{Z}$.

- **Predictive Equality:** A predictor satisfies predictive equality if

$$P\big(\hat{Y} = 1 \mid Y = 0, Z = z\big) = P\big(\hat{Y} = 1 \mid Y = 0\big) \quad \forall z.$$

- **Individual Fairness:** A predictor $f \colon \mathcal{X} \to \hat{\mathcal{Y}}$ is individually fair if

$$d_{\hat{\mathcal{Y}}}\big(f(x), f(x')\big) \ \le \ d_{\mathcal{X}}(x, x') \quad \forall\, x, x' \in \mathcal{X}.$$

- **Calibration within Groups:** A score $\hat{p}(X)$ is calibrated within groups if

$$P\big(Y = 1 \mid \hat{p}(X) = s,\ Z = z\big) = s \quad \forall\, s \in [0, 1],\ z \in \mathcal{Z}.$$

- **Counterfactual Fairness:** Given a structural causal model with background $U$,

$$P(\hat{Y}_{A \leftarrow a}(U) = y|X = x, Z = z) = P(\hat{Y}_{A \leftarrow a'}(U) = y|X = x, Z = z), \quad \forall z, z' \in \mathcal{Z}, \quad \forall x \in \mathcal{X}.$$

# B  APPENDIX TO SECTION 2

## B.1  PROOF OF LEMMA 2.4

*Proof.* Assume $\mathbb{E}[f(\hat{Y})] = 0$ and $\mathbb{E}[g(Z)] = 0$, we have the following sequence of equalities and inequalities:

$$
\begin{aligned}
\frac{\left|\langle f(\hat{Y}), g(Z)\rangle_{L^2}\right|}{\|f\|_{L^\infty}\|g\|_{L^\infty}} &= \frac{1}{\|f\|_{L^\infty}\|g\|_{L^\infty}} \left|\int_{\mathcal{Y}\times\mathcal{Z}} f(y)g(z)\, d\mathbb{P}_{(\hat{Y},Z)}(y,z)\right| \\
&= \frac{1}{\|f\|_{L^\infty}\|g\|_{L^\infty}} \left|\int_{\mathcal{Y}\times\mathcal{Z}} f(y)g(z)\, d(\mathbb{P}_{(\hat{Y},Z)} - \mathbb{P}_{\hat{Y}}\otimes\mathbb{P}_Z)(y,z)\right| \\
&\leq \frac{1}{\|f\|_{L^\infty}\|g\|_{L^\infty}} \int_{\mathcal{Y}\times\mathcal{Z}} |f(y)g(z)|\, d|\mathbb{P}_{(\hat{Y},Z)} - \mathbb{P}_{\hat{Y}}\otimes\mathbb{P}_Z|(y,z) \\
&\leq \int_{\mathcal{Y}\times\mathcal{Z}} d|\mathbb{P}_{(\hat{Y},Z)} - \mathbb{P}_{\hat{Y}}\otimes\mathbb{P}_Z|(y,z) \\
&= 2\,\|\mathbb{P}_{(\hat{Y},Z)} - \mathbb{P}_{\hat{Y}}\otimes\mathbb{P}_Z\|_{\mathrm{TV}} \\
&\leq 2\sqrt{\frac{1}{2}D_{\mathrm{KL}}\left(\mathbb{P}_{(\hat{Y},Z)} \,\|\, \mathbb{P}_{\hat{Y}}\otimes\mathbb{P}_Z\right)} \\
&= \sqrt{2\,I(\hat{Y};Z)}.
\end{aligned}
$$

Here, the forth line follows from $|f(y)g(z)| \leq \|f\|_\infty\|g\|_\infty$, the fifth from definition of TV norm, sixth from Pinsker's inequality, and the last from definition of mutual information. We are done. $\square$

## B.2  EXAMPLES OF LEMMA 2.4

On direct application or corollary of the above result is that one can effectively upper bound the covariance between $f(\hat{Y})$ and $g(Z)$ for any bounded $f$ and $g$: If $I(\hat{Y};Z) \leq C$, then for any bounded $f, g$,

$$
\left|\mathrm{Cov}(f(\hat{Y}), g(Z))\right| \leq \|f(\hat{Y})\|_{L^\infty}\|g(Z)\|_{L^\infty}\sqrt{2C}.
$$

That is, a bound on mutual information uniformly controls covariance for *all* bounded queries. Below, we provide two concrete examples:

*Example* B.1 (Binary Indicator). Let $f(\hat{Y}) = \mathbf{1}\{h(\hat{Y}) = 1\}$ and $g(Z) = \mathbf{1}\{a(Z) = 1\}$ for (possibly randomized) binary classifiers $h, a$. Then

$$
\mathrm{Cov}\big(\mathbf{1}\{h(\hat{Y}) = 1\},\, \mathbf{1}\{a(Z) = 1\}\big) = \mathbb{P}\big(h(\hat{Y}) = 1,\, a(Z) = 1\big) - \mathbb{P}\big(h(\hat{Y}) = 1\big)\,\mathbb{P}\big(a(Z) = 1\big).
$$

Moreover, by Theorem

$$
\left|\mathrm{Cov}(f(\hat{Y}), g(Z))\right| \leq \|f\|_\infty\|g\|_\infty\sqrt{2\,I(\hat{Y};Z)}.
$$

Since $\|f\|_\infty = \|g\|_\infty = 1$ for indicators, we obtain the explicit bound

$$
\left|\mathbb{P}(h(\hat{Y}) = 1,\, a(Z) = 1) - \mathbb{P}(h(\hat{Y}) = 1)\mathbb{P}(a(Z) = 1)\right| \leq \sqrt{2\,I(\hat{Y};Z)}.
$$

In Lemma 2.4, we assume the knowledge of the population mutual information. In practice, one would apply the sample or empirical estimation of mutual information to upper bound the sample or empirical estimation of the covariance between $f(\hat{Y})$ and $g(Z)$ for any (essentially) bounded $f$ and $g$. The next result applies concentration inequality to show the empirical estimation error and corresponding probability guarantee, given a sample size, so that together with Lemma 2.4, we can provide a uniform covariance control guarantee only based on sample estimations.

*Example* B.2 (1-dimensional Covariance Control). Given i.i.d. samples $\{(\hat{Y}_t, Z_t)\}_{t=1}^n$ and bounded $f, g$ with $M_f := \|f(\hat{Y})\|_\infty$ and $M_g := \|g(Z)\|_\infty$, define

$$\widehat{\text{Cov}}_n(f, g) := \frac{1}{n}\sum_{t=1}^n f(\hat{Y}_t)g(Z_t) - \Big(\frac{1}{n}\sum_{t=1}^n f(\hat{Y}_t)\Big)\Big(\frac{1}{n}\sum_{t=1}^n g(Z_t)\Big).$$

Then, for any $\delta \in (0, 1)$, with probability at least $1 - \delta$,

$$\big|\widehat{\text{Cov}}_n(f, g) - \text{Cov}(f(\hat{Y}), g(Z))\big| \;\leq\; M_f M_g \sqrt{\frac{2\log(6/\delta)}{n}}. \tag{11}$$

*Proof.* Write

$$\widehat{\text{Cov}}_n(f, g) - \text{Cov}(f(\hat{Y}), g(Z)) = \underbrace{\Big(\tfrac{1}{n}\sum_{t=1}^n f_t g_t - \mathbb{E}[fg]\Big)}_{(A)} - \underbrace{\big(\overline{f}_n \overline{g}_n - \mathbb{E}f\,\mathbb{E}g\big)}_{(B)},$$

where $f_t := f(\hat{Y}_t)$, $g_t := g(Z_t)$, $\overline{f}_n := n^{-1}\sum_t f_t$, $\overline{g}_n := n^{-1}\sum_t g_t$. By Hoeffding's inequality, since $|f_t g_t| \leq M_f M_g$,

$$\mathbb{P}\Bigg(|(A)| \geq M_f M_g \sqrt{\frac{2\log(6/\delta)}{n}}\Bigg) \leq \frac{\delta}{3}.$$

Also,
$$\big|(B)\big| \leq |\overline{f}_n - \mathbb{E}f|\,|\overline{g}_n| + |\mathbb{E}f|\,|\overline{g}_n - \mathbb{E}g| \leq M_g\,|\overline{f}_n - \mathbb{E}f| + M_f\,|\overline{g}_n - \mathbb{E}g|.$$

Applying Hoeffding's inequality again to each mean (since $|f_t - \mathbb{E}f| \leq 2M_f$, $|g_t - \mathbb{E}g| \leq 2M_g$) and a union bound yields, with probability at least $1 - \frac{2\delta}{3}$,

$$|\overline{f}_n - \mathbb{E}f| \leq M_f \sqrt{\frac{2\log(6/\delta)}{n}}, \qquad |\overline{g}_n - \mathbb{E}g| \leq M_g \sqrt{\frac{2\log(6/\delta)}{n}}.$$

Combining the three events and simplifying constants gives equation 11. $\qquad\square$

Together with the population bound $|\text{Cov}(f(\hat{Y}), g(Z))| \leq \|f\|_\infty \|g\|_\infty \sqrt{2\,I(\hat{Y}; Z)}$, the empirical covariance is controlled by (i) structural dependence via mutual information and (ii) sampling noise at the usual $n^{-1/2}$ rate:

$$|Cov(f, g)| \leq |Cov(f, g) - \widehat{Cov}(f, g)| + |\widehat{Cov}(f, g)|$$

$$\leq \|f\|_\infty \|g\|_\infty \Big(\sqrt{\frac{2\log(6/\delta)}{n}} + \sqrt{2I(\hat{Y}; Z)}\,\Big)$$

with probability at least $1 - \delta$.

## B.3 COUNTER-EXAMPLE FOR CORRELATION

*Example* B.3 (Why not correlation?). Let $Y$ be a Bernoulli variable with $P(Y = 0) = \epsilon$ (small) and $P(Y = 1) = 1 - \epsilon$. Set $Z = Y$ (perfectly correlated). Then

$$I(Y; Z) = H(Y) = -\epsilon \ln \epsilon - (1 - \epsilon)\ln(1 - \epsilon),$$

which becomes approximately $\epsilon|\ln \epsilon|$ when $\epsilon \to 0$. Now choose centered, bounded functions

$$f(Y) = \frac{\mathbf{1}_{Y=0} - \epsilon}{\sqrt{\epsilon(1-\epsilon)}}, \qquad g(Z) = \frac{\mathbf{1}_{Z=0} - \epsilon}{\sqrt{\epsilon(1-\epsilon)}}.$$

By construction $E[f] = E[g] = 0$, and one checks

$$\text{Cov}(f(Y), g(Z)) = \frac{E[(\mathbf{1}_{Y=0} - \epsilon)^2]}{\epsilon(1-\epsilon)} = \frac{\epsilon(1-\epsilon)}{\epsilon(1-\epsilon)} = 1.$$

In particular, with $L^2$ normalization we get $\frac{|\text{Cov}(f,g)|}{|f|_2|g|_2} = 1$. But $\sqrt{2I(Y; Z)} \approx \sqrt{2\epsilon|\ln \epsilon|} \to 0$ as $\epsilon \to 0$. Thus the $L^2$-normalized covariance can be much larger than $\sqrt{2I}$, violating any bound of the form $|\text{Cov}(f, g)|/(|f|_2|g|_2) \leq \sqrt{2I(Y; Z)}$.

### B.4 Proof of Proposition 2.3

*Proof.* ($\Longrightarrow$) By Theorem 2.5,

$$\sup_{f,g} \frac{\mathbb{E}_Y\big[\langle f(\hat{Y}_Y),\, g(Z_Y)\rangle\big]}{\|f\|_\infty\,\|g\|_\infty} \;\leq\; \sqrt{2\,\mathbb{E}\big[I(\hat{Y};Z\mid Y)\big]} \;=\; 0.$$

Hence, for almost every $y$ and for all bounded $f, g$ with $\mathbb{E}[f(\hat{Y}_y)] = \mathbb{E}[g(Z_y)] = 0$,

$$\mathbb{E}\big[f(\hat{Y})\,g(Z)\ \big|\ Y=y\big] = 0.$$

Equivalently, every pair of zero-mean bounded functions of $\hat{Y}$ and $Z$ is uncorrelated conditional on $Y = y$. By the functional-moment characterization of independence (since bounded functions separate probability measures), this implies that for almost every $y$,

$$P_{\hat{Y},Z|Y=y} \;=\; P_{\hat{Y}|Y=y} \otimes P_{Z|Y=y},$$

i.e. $\hat{Y} \perp Z \mid Y$.

($\Longleftarrow$) By definition,

$$I(\hat{Y};Z\mid Y) = \mathbb{E}_Y\Big[D_{\mathrm{KL}}\big(P_{\hat{Y},Z|Y}\,\big\|\,P_{\hat{Y}|Y}\,P_{Z|Y}\big)\Big].$$

But conditional independence means $P_{\hat{Y},Z|Y=y} \;=\; P_{\hat{Y}|Y=y}\,P_{Z|Y=y}$ for almost every $y$, and $D_{\mathrm{KL}}(P\|P) = 0$. Hence each inner KL vanishes and therefore $I(\hat{Y};Z\mid Y) = 0$. $\qquad\square$

### B.5 Proof of Proposition 2.1

*Proof.* The feasible set $\{f:\ I(f(X,Z);Z\mid Y) \leq v\}$ is monotonic in set inclusion with $v$. $\qquad\square$

### B.6 Proof of Theorem 2.2

*Proof. Step 1 (W.l.o.g. reduction to a uniform selector).* Any standard Borel probability space $(\mathcal{B}, \mu)$ is isomorphic to the unit interval $([0,1], \lambda)$ with Lebesgue measure $\lambda$ (See, for example, (Kallenberg, 1997, Theorem 1.8)). This means any revealed randomization scheme can be realized, without loss of generality, by a selector $B \sim \mathrm{Unif}[0,1]$ and a jointly measurable $F : \mathcal{X} \times \mathcal{Z} \times [0,1] \to \mathcal{U}$. We assume this simplified representation for the remainder of the proof.

*Step 2 (Average identity).* Let $U_b := f_b(X,Z)$ and $\widetilde{U} := (B, U_B)$ with $B \sim \mathrm{Unif}[0,1]$. The assumption that all spaces are Polish with Borel sigma-algebra, which guarantees the existence of regular conditional distributions. Therefore, it follows from chain rule for mutual information that

$$I(\widetilde{U};Y) = I(B, U_B;Y) = I(B;Y) + I(U_B;Y\mid B)$$
$$I(\widetilde{U};Z\mid Y) = I(B, U_B;Z\mid Y) = I(B;Z\mid Y) + I(U_B;Z\mid Y,B)$$

By assumption, $B$ is independent of $(X,Y,Z)$, which implies $B \perp Y$ and $B \perp Z \mid Y$. Therefore, $I(B;Y) = 0$ and $I(B;Z\mid Y) = 0$. The chain rules now simplify to:

$$I(\widetilde{U};Y) = I(U_B;Y\mid B) = \int_0^1 I(U_b;Y)\,db$$

$$I(\widetilde{U};Z\mid Y) = I(U_B;Z\mid Y,B) = \int_0^1 I(U_b;Z\mid Y)\,db$$

The last equality in each line from disintegration that $b \mapsto I(U_b;Y)$ and $b \mapsto I(U_b;Z\mid Y)$ are Borel measurable functions. Thus, every randomization point on the $(u,v)$ plane is the average of some deterministic point parametrized by the realization $B = b$:

$$\big(I(\widetilde{U};Z\mid Y),\, I(\widetilde{U};Y)\big) = \int_0^1 \big(v_{f_b},\, u_{f_b}\big)\,db. \tag{12}$$

*Step 3 (conv$(\mathcal{S}_{\text{det}}) \subseteq \mathcal{S}_{\text{rand}}$).* We show that any finite convex combination $y = \sum_{j=1}^{J} \alpha_j (v_{f_j}, u_{f_j})$ is in $\mathcal{S}_{\text{rand}}$, so that conv$(\mathcal{S}_{\text{det}}) \subseteq \mathcal{S}_{\text{rand}}$. We construct a time-sharing scheme by partitioning the interval $[0,1]$ into disjoint intervals $A_j$ with lengths $|A_j| = \alpha_j$. We then define the measurable map $F$ such that $f_b(x,z) := f_j(x,z)$ for all $b \in A_j$. Applying the integral identity equation 12 to this scheme:

$$\int_0^1 (v_{f_b}, u_{f_b})\, db = \sum_{j=1}^{J} \int_{A_j} (v_{f_j}, u_{f_j})\, db = \sum_{j=1}^{J} |A_j| (v_{f_j}, u_{f_j}) = y.$$

Since $y$ is achieved by a valid scheme, $y \in \mathcal{S}_{\text{rand}}$. Thus, conv$(\mathcal{S}_{\text{det}}) \subseteq \mathcal{S}_{\text{rand}}$.

*Step 4 ($\mathcal{S}_{\text{rand}} \subseteq \overline{\text{conv}}(\mathcal{S}_{\text{det}})$).* Let $y$ be any point in $\mathcal{S}_{\text{rand}}$ achieved by some randomized scheme $(B, \tilde{U}_B)$. By equation 12, $y = \int_0^1 s(b)\, db$, where $s(b) := (v_{f_b}, u_{f_b}) \in \mathcal{S}_{\text{det}}$. The finiteness of entropy and mutual information ensure $s(b)$ is bounded, so $s \in L^1([0,1]; \mathbb{R}^2)$. Since simple functions are dense in $L^1$, for any $\varepsilon > 0$, we can find a finite partition $\{E_k\}_{k=1}^{m}$ of $[0,1]$ and points $b_k \in E_k$ such that the finite convex combination $y_{\text{simple}} := \sum_{k=1}^{m} |E_k| s(b_k)$ satisfies $\|y - y_{\text{simple}}\|_{L^1} < \varepsilon$. Since each $s(b_k) \in \mathcal{S}_{\text{det}}$, each $y_{\text{simple}}$ is in conv$(\mathcal{S}_{\text{det}})$. Because $y$ is a limit point of elements from conv$(\mathcal{S}_{\text{det}})$, it must lie in its closure. Therefore, $\mathcal{S}_{\text{rand}} \subseteq \overline{\text{conv}}(\mathcal{S}_{\text{det}})$.

By Step 3 and 4, we have $\overline{\text{conv}}(\mathcal{S}_{\text{det}}) \subseteq \mathcal{S}_{\text{rand}} \subseteq \overline{\text{conv}}(\mathcal{S}_{\text{det}})$. Finally, since the supremum of a set is equal to the supremum of its closure, the upper frontiers of the two sets are identical: $U_{\mathcal{S}_{\text{rand}}}(v) = U_{\overline{\text{conv}}(\mathcal{S}_{\text{det}})}(v)$ for all $v \geq 0$. That completes the proof. $\square$

### B.7 PROOF OF LEMMA 2.6

*Proof.* Fix a generic predictor $U$, it follows from the chain rule for mutual information that $I(U;(Y,Z)) = I(U;Y) + I(U;Z \mid Y) = u + v$ by definition. The upper bound $v \leq H(Z \mid Y)$ follows from the fact that mutual information is upper bounded by the smaller entropy of the two individual random variables, which still holds true when conditioned on $Y$ here. The upper bound $u \leq I((X,Z);Y)$ follows from data processing inequality for Markov chains $U - (X,Z) - Y$. The upper bound

$$u + v \leq I((X,Z);(Y,Z)) = I((X,Z);Y) + I((X,Z);Z \mid Y) = I((X,Z);Y) + H(Z \mid Y)$$

follows from the chain rule for mutual information and the data processing inequality for Markov chain $U - (X,Z) - (Y,Z)$ respectively. Finally, $I(U;Y) \leq H(Y) \leq \log|\mathcal{Y}| < \infty$ due to the assumption that the outcome space $\mathcal{Y}$ has finite cardinality. That completes the proof. $\square$

### B.8 PROOF OF LEMMA 2.7

*Proof.* The assumption that the spaces are Polish ensures they are standard Borel spaces, which guarantees the existence of regular conditional probability distributions, which validates the steps that follow.

For a fixed $z$, the predictor $f(X,z)$ is a measurable function of $X$. By the assumption $X \perp Z \mid Y$ and data processing inequality, we have $f(X,z)$ is also conditionally independent of $Z$ given $Y$. Therefore, for any Borel set $A \subset \mathcal{U}$, we have for $P_{Y,Z}$-almost all $(y,z)$:

$$\mathbb{P}\big(f(X,z) \in A \mid Y = y, Z = z\big) = \mathbb{P}\big(f(X,z) \in A \mid Y = y\big).$$

Moreover, by the assumption $I(U;Z \mid Y) = 0$, we have $U \perp Z \mid Y$. Hence,

$$\mathbb{P}\big(U \in A \mid Y = y, Z = z\big) = \mathbb{P}\big(U \in A \mid Y = y\big),$$

for $P_{Y,Z}$-almost all $(y,z)$ By the definition of the predictor, $U = f(X,Z)$, their conditional distributions given $(Y,Z)$ are identical:

$$\mathbb{P}\big(U \in A \mid Y = y, Z = z\big) = \mathbb{P}\big(f(X,z) \in A \mid Y = y, Z = z\big).$$

Now, combining these three equalities together, we have

$$\mathbb{P}\big(f(X,z) \in A \mid Y = y\big) = \mathbb{P}\big(U \in A \mid Y = y\big).$$

for $P_{Y,Z}$-almost all $(y,z)$.

It remains to show that the statements hold "for a.e. $z_0$, for a.e. $y$". To that end, let $\mathcal{A}$ be a $\pi$-system, the set of finite intersection of rational radii balls with centers on a dense set of $\mathcal{U}$, for the Borel $\sigma$-algebra $\mathcal{B}(\mathcal{U})$. For each $A \in \mathcal{A}$, let $N_A = \{(y, z) \mid \mathbb{P}(f(X, z) \in A \mid y) \neq \mathbb{P}(U \in A \mid y)\}$. We know $P_{Y,Z}(N_A) = 0$. Since $\mathcal{U}$ is Polish, the set $N = \bigcup_{A \in \mathcal{A}} N_A$ is a countable union of null sets. So it is also a $P_{Y,Z}$-null set.

Now, it follows from Fubini's theorem that $Z_N = \{z \mid P_Y(\{y \mid (y, z) \in N\}) > 0\}$ is a $P_Z$-null set. Let $Z_0 := \mathcal{Z} \setminus Z_N$, we have ($P_Z(Z_0) = 1$). By construction, for any $z_0 \in Z_0$, the $y$-slice $\{y \mid (y, z_0) \in N\}$ is a $P_Y$-null set. That implies, for any $z_0 \in Z_0$, the equality $\mathbb{P}(f(X, z_0) \in A \mid y) = \mathbb{P}(U \in A \mid y)$ holds for all $A \in \mathcal{A}$ and for $P_Y$-almost all $y$.

By Dynkin's $\pi$-$\lambda$ Theorem, since the two conditional measures $\mathcal{L}(f(X, z_0) \mid Y = y)$ and $\mathcal{L}(U \mid Y = y)$ agree on $\mathcal{A}$, they agree on the entire Borel $\sigma$-algebra. Thus, $\mathcal{L}(f(X, z_0) \mid Y = y) = \mathcal{L}(U \mid Y = y)$ for $P_Y$-a.e. $y$.

Integrating this conditional law against the measure $P_Y(dy)$ shows that the joint laws are equal: $\mathcal{L}(f(X, z_0), Y) = \mathcal{L}(U, Y)$. Since mutual information is a function of the joint law, their mutual information must also be identical. $\qquad\square$

### B.9 PROOF OF THEOREM 2.8

*Proof.* (i) Since $U = g(X)$ satisfies $I(U; Z \mid Y) = 0$ by assumption $X \perp Z \mid Y$ for all $g$, we have

$$\sup \Big\{ I(U; Y) : \ U = f(X, Z), \ I(U; Z \mid Y) = 0 \Big\} \geq u_X^\star.$$

On the other hand, let $U = f(X, Z)$ satisfy $I(U; Z \mid Y) = 0$. By Lemma 2.7, pick arbitrary $z_0 \in Z_0$ with $I(f(X, z_0); Y) = I(U; Y)$, we have

$$I(U; Y) \ = \ I\big(f(X, z_0); Y\big) \ \leq \ \sup_g I\big(g(X); Y\big) = u_X^\star.$$

That completes our proof for (i).

(ii) Now, if $I(Z; Y \mid X) > 0$, then by the chain rule, we have

$$I(X, Z; Y) = I(X; Y) + I(Z; Y \mid X) \ > \ I(X; Y).$$

Since $u_X^\star \leq I(X; Y)$ by data processing and $u_{XZ}^\star \geq I(X, Z; Y)$ (take $f(x, z) = (x, z)$, which is measurable into a Polish space), we have $u_{XZ}^\star > u_X^\star$. Thus any $U$ with $I(U; Y) > u_X^\star$ cannot satisfy the constraint $I(U; Z \mid Y) = 0$ from part (i), so necessarily $I(U; Z \mid Y) > 0$. $\qquad\square$

---

**Algorithm 1** CMI-regularized training (surrogate for EO) with grad-norm balancing

---

**Require:** Dataset $\mathcal{D} = \{(x_i, y_i, z_i)\}$, model $f_\theta = g_\theta \circ \phi_\theta$, mixing weight $\lambda \in [0, 1]$, optimizer (Adam), $\epsilon > 0$
**Ensure:** Trained parameters $\theta$
1: **for** epoch $= 1, \ldots, T$ **do**
2:    **for** mini-batch $\mathcal{B} = \{(x, y, z)\}$ **do**
3:       **Compose input:** $x_{\text{in}} \leftarrow (x, z)$  (if using sensitive input; else $x_{\text{in}} = x$)
4:       **Forward:** $(l, h) \leftarrow (g_\theta(\phi_\theta(x_{\text{in}})), \phi_\theta(x_{\text{in}}))$
5:       **Raw losses:**
6:          $\mathcal{L}_{\text{task}} \leftarrow \text{CE}(l, y)$
7:          $\widehat{I}_{\text{CMI}} \leftarrow \text{SOFTCMI}(l, z, y)$   (computed from $\text{softmax}(l)$)
8:       **Grad-norms on features (detached):**
9:          $n_{\text{task}} \leftarrow \mathbb{E}_i \left\| \partial_{h_i} \mathcal{L}_{\text{task}} \right\|_2$ (stop-grad)
10:         $n_{\text{cmi}} \leftarrow \mathbb{E}_i \left\| \partial_{h_i} \widehat{I}_{\text{CMI}} \right\|_2$ (stop-grad)
11:       **Balanced objective:** $\mathcal{L}_{\text{final}} \leftarrow (1 - \lambda) \dfrac{\mathcal{L}_{\text{task}}}{n_{\text{task}} + \epsilon} + \lambda \dfrac{\widehat{I}_{\text{CMI}}}{n_{\text{cmi}} + \epsilon}$
12:       **Update:** take an optimizer step on $\nabla_\theta \mathcal{L}_{\text{final}}$ (Adam)
13:    **end for**
14: **end for**
15: **return** $\theta$

---

## C   Appendix of Section 3

Based on the theoretical insights from Section 2, we now leverage a sample estimator of conditional mutual information (CMI) to steer the learned predictor toward the separation-utility Pareto frontier.

Given a dataset $\mathcal{D} = \{(x_i, y_i, z_i)\}_{i=1}^N$, we train a model $f_\theta$ consisting of a feature extractor $h = \phi(x, z)$ and a classifier $g(h)$. To ensure optimization stability, particularly when balancing competing objectives with different gradient scales, we employ gradient normalization. We minimize a dynamic objective where each term is scaled by the inverse norm of its gradient with respect to the features $h$:

$$\mathcal{L}_{\text{total}} = (1 - \gamma) \frac{\mathcal{L}_{\text{task}}}{\|\nabla_h \mathcal{L}_{\text{task}}\|} + \gamma \frac{\widehat{I}_{\text{CMI}}}{\|\nabla_h \widehat{I}_{\text{CMI}}\|}, \tag{13}$$

where $\gamma \in [0, 1]$ controls the trade-off. This normalization ensures that both the utility signal and the fairness penalty contribute equally to the update magnitude, preventing one from dominating the other due to arbitrary scaling.

### C.1   Differentiable Soft-Plug-in Estimator

To backpropagate through $I(U; Z \mid Y)$, we implement a direct sample estimator. Let $p_i = \text{softmax}(f_\theta(x_i)) \in \Delta^{|\mathcal{Y}|}$ be the predicted probability vector for sample $i$.

The estimator $\widehat{I}_{\text{CMI}}$ is computed on a mini-batch $B$ as follows:

1. *Stratification:* Partition the batch indices by target class: $\mathcal{I}_y = \{i \in B : y_i = y\}$.
2. *Soft Joint Distribution:* For each class $y$ and sensitive group $z$, compute the average soft prediction:

$$\hat{P}(U|y, z) = \frac{1}{|\{i \in \mathcal{I}_y : z_i = z\}|} \sum_{i \in \mathcal{I}_y, z_i = z} p_i$$

3. *Divergence Aggregation:* The conditional MI is computed as the weighted sum of KL divergences between the sensitive-conditional distributions and the marginal distribution within each class $y$:

$$\widehat{I}_{\text{CMI}} = \sum_y \frac{|\mathcal{I}_y|}{|B|} \sum_z \hat{P}(z|y) \, D_{\text{KL}}\big(\hat{P}(U|y, z) \, \| \, \hat{P}(U|y)\big)$$

## C.2 STATISTICAL CONSISTENCY AND EXPLICIT BIAS ANALYSIS

A critical feature of the plug-in CMI estimator is its statistical guaranties on finite batches that can be monitored during training. Unlike adversarial discriminators or variational bound estimators that can under-estimate dependence, the sample estimator is **biased upwards**, effectively serving as a conservative upper bound on the true separation violation.

We provide a rigorous characterization of this bias and the convergence rate below.

**Proposition C.1** (Bias and Concentration of Sample CMI). *Let $(U, Y, Z)$ take values in discrete spaces with cardinalities $K_U, K_Y, K_Z$. Let $\widehat{I}_B$ be the sample estimator of $I(U; Z \mid Y)$ computed on $|B|$ i.i.d. samples. Assume $P(Y = y) \geq p_{\min} > 0$ and conditional probabilities $P(u, z \mid y) \geq q_{\min} > 0$.*

*1. **Asymptotic Bias:** As $|B| \to \infty$, the estimator is positively biased:*

$$\mathbb{E}[\widehat{I}_B] - I(U; Z \mid Y) = \frac{K_Y(K_U - 1)(K_Z - 1)}{2|B|} + O(|B|^{-2}).$$

*2. **Concentration:** For any $\delta \in (0, 1)$, if $|B|$ is sufficiently large, then with probability $\geq 1 - \delta$:*

$$|\widehat{I}_B - \mathbb{E}[\widehat{I}_B]| \leq C\sqrt{\frac{\log(2/\delta)}{|B|}}, \tag{14}$$

*where $C = C(K_U, K_Y, K_Z, p_{min}, q_{min})$.*

*Proof.* Write the plug-in estimator as a stratified average

$$\widehat{I}_B = \sum_{y \in \mathcal{Y}} \frac{B_y}{|B|} \widehat{I}_y, \qquad \widehat{I}_y := \widehat{I}(U; Z \mid Y = y),$$

where $B_y := |\{i : Y_i = y\}|$ is the number of samples with label $y$, and $\widehat{I}_y$ is the ordinary empirical mutual information computed from those $B_y$ samples.

**Part (1): Asymptotic bias.** Fix $y \in \mathcal{Y}$ and condition on $\{B_y = n\}$ with $n \geq 1$. On this event, $\widehat{I}_y$ is the standard empirical MI estimator from $n$ i.i.d. samples drawn from the conditional distribution $P_{UZ|Y=y}$. Under the full-support assumption ($P(u, z|y) \geq q_{\min} > 0$), the Miller–Madow expansion for the empirical MI bias (see Paninski (2003)) yields:

$$\mathbb{E}[\widehat{I}_y \mid B_y = n] = I(U; Z \mid Y = y) + \frac{(K_U - 1)(K_Z - 1)}{2n} + O(n^{-2}). \tag{15}$$

Multiply equation 15 by $B_y/|B|$ and take expectations over $B_y$:

$$\mathbb{E}\left[\frac{B_y}{|B|}\widehat{I}_y\right] = \mathbb{E}\left[\frac{B_y}{|B|}\right] I(U; Z \mid Y = y) + \frac{(K_U - 1)(K_Z - 1)}{2|B|}\mathbb{E}[\mathbb{1}_{\{B_y \geq 1\}}] + R_y,$$

where the remainder satisfies

$$|R_y| \leq \mathbb{E}\left[\frac{B_y}{|B|}O(B_y^{-2})\mathbb{1}_{\{B_y \geq 1\}}\right] = O\left(\frac{1}{|B|}\mathbb{E}\left[\frac{1}{B_y}\mathbb{1}_{\{B_y \geq 1\}}\right]\right).$$

Since $B_y \sim \text{Binomial}(|B|, p_y)$ with $p_y := P(Y = y) \geq p_{\min}$, a Chernoff bound gives $\mathbb{P}(B_y \leq \frac{1}{2}|B|p_y) \leq e^{-c|B|}$ for some $c = c(p_{\min}) > 0$. The inverse moment is dominated by the typical set:

$$\mathbb{E}\left[\frac{1}{B_y}\mathbb{1}_{\{B_y \geq 1\}}\right] \leq \frac{2}{|B|p_y} + \mathbb{P}(B_y \leq \frac{1}{2}|B|p_y) = O(|B|^{-1}).$$

Therefore, $R_y = O(|B|^{-2})$. Also, $\mathbb{E}[B_y/|B|] = p_y$ and $\mathbb{E}[\mathbb{1}_{\{B_y \geq 1\}}] = 1 - (1 - p_y)^{|B|} = 1 + O(e^{-c|B|})$. Summing over $y \in \mathcal{Y}$ gives

$$\mathbb{E}[\widehat{I}_B] = \sum_y p_y I(U; Z \mid Y = y) + \frac{K_Y(K_U - 1)(K_Z - 1)}{2|B|} + O(|B|^{-2}),$$

and $\sum_y p_y I(U; Z \mid Y = y) = I(U; Z \mid Y)$. That proves part (1).

**Part (2): Concentration.** We use a high-probability "good" event to avoid empty/near-empty strata (conditioned on $\{Y = y\}$) which cause instability in the entropy estimates. Define

$$\mathcal{G} := \left\{ \forall y : \; B_y \geq \tfrac{1}{2}|B|p_{\min} \right\} \cap \left\{ \forall(u, z, y) : \; N_{u,z,y} \geq \tfrac{1}{2}|B|\, p_{\min}q_{\min} \right\},$$

where $N_{u,z,y}$ is the count of samples equal to $(u, z, y)$. By Chernoff bounds and a union bound over all strata and outcomes, there exists an absolute constant $c_0 > 0$ such that:

$$\epsilon_B := \mathbb{P}(\mathcal{G}^c) \leq (K_Y + K_Y K_U K_Z) \exp\big( - c_0 |B| p_{\min}q_{\min} \big). \tag{16}$$

For sufficiently large $|B|$, $\epsilon_B \leq \delta/2$.

Next, we establish the bounded difference property on $\mathcal{G}$. Let $F(D) := \widehat{I}_B$ viewed as a function of the dataset $D$. Consider two datasets $D, D'$ differing by exactly one sample. This change affects at most two strata (if the $Y$-label changes). On $\mathcal{G}$, all empirical probabilities are bounded below by $q_{\min}/2$. The function $h(t) = -t \log t$ has derivative $|h'(t)| = |1 + \log t| \leq 1 + \log(2/q_{\min}) =: L_0$ on $[q_{\min}/2, 1]$. Thus, the entropy functional is $L_0$-Lipschitz with respect to the $\ell_1$ norm. Changing one sample in stratum $y$ changes the empirical distribution by at most $2/B_y$. Consequently, the change in the local MI estimate $\widehat{I}_y$ is bounded:

$$|\Delta \widehat{I}_y| \leq 3L_0 \|\Delta \widehat{P}_{UZ|y}\|_1 \leq \frac{6L_0}{B_y}.$$

The total change in the weighted sum $\widehat{I}_B = \sum \frac{B_y}{|B|} \widehat{I}_y$ involves changes to the weights $B_y/|B|$ and the values $\widehat{I}_y$. Since $B_y \geq \frac{1}{2}|B|p_{\min}$ on $\mathcal{G}$, and $\widehat{I}_y$ is globally bounded by $M_{\max} = \log(K_U K_Z)$, there exists a constant $c := 2M_{\max} + 12L_0$ such that

$$|F(D) - F(D')| \leq \frac{c}{|B|} \quad \forall D, D' \in \mathcal{G} \text{ with } d_H(D, D') = 1.$$

Here, $d_H$ is the Hamming distance. To apply concentration bounds rigorously despite the conditioning, we consider a Lipschitz extension. Since $F$ satisfies the bounded difference property with constant $c/|B|$ on $\mathcal{G}$, there exists an extension $F^\star$ defined on the entire domain $\mathcal{Z}^{|B|}$ that coincides with $F$ on $\mathcal{G}$ and preserves the bounded differences property globally. Explicitly, we use the McShane-Whitney extension with Hamming distance $d_H$:

$$F^\star(D) := \inf_{D' \in \mathcal{G}} \left( F(D') + \frac{c}{|B|} d_H(D, D') \right).$$

By construction, $F^\star(D) = F(D)$ whenever $D \in \mathcal{G}$, and $F^\star$ satisfies the bounded difference condition with constant $c/|B|$ everywhere. Applying McDiarmid's inequality to $F^\star$:

$$\mathbb{P}\big(|F^\star(D) - \mathbb{E}[F^\star]| \geq t\big) \leq 2 \exp\left( -\frac{2|B|t^2}{c^2} \right).$$

Now, since $\{F \neq F^\star\} \subseteq \mathcal{G}^c$, and the global range is bounded by $M_{\max}$, we have

$$|\mathbb{E}[F^\star] - \mathbb{E}[F]| \leq M_{\max}\mathbb{P}(\mathcal{G}^c) = M_{\max}\epsilon_B.$$

Thus, for any $t > 0$:

$$\mathbb{P}\big(|F - \mathbb{E}[F]| \geq t + M_{\max}\epsilon_B\big) \leq 2 \exp\left( -\frac{2|B|t^2}{c^2} \right) + \epsilon_B.$$

Setting $t = c\sqrt{\frac{\log(4/\delta)}{2|B|}}$, the first term becomes $\leq \delta/2$. For $|B|$ large, $\epsilon_B \leq \delta/2$ and the exponentially decaying bias term $M_{\max}\epsilon_B$ is negligible compared to the concentration term (specifically, $M_{\max}\epsilon_B \leq \frac{c}{\sqrt{2}}\sqrt{\frac{\log(2/\delta)}{|B|}}$). Summing the terms and simplifying, we define the constant $C := \sqrt{2}c = 2\sqrt{2}(M_{\max} + 6L_0)$. This yields the final bound:

$$\left| \widehat{I}_B - \mathbb{E}[\widehat{I}_B] \right| \leq C\sqrt{\frac{\log(2/\delta)}{|B|}} \quad \text{with probability } \geq 1 - \delta.$$

This completes the proof of Part (2). $\qquad\square$

| Dataset | # Samples | Input Features ($X$) | Target ($Y$) | Sensitive ($Z$) |
|---|---|---|---|---|
| Adult | 48,842 | Tabular (14 numeric/cat.) | Income $\geq$ 50K | Gender (M/F) |
| COMPAS | 6,172 | Tabular (12 numeric/cat.) | Recidivism $\leq$ 2 yrs | Race (Black/White) |
| Bank | 45,211 | Tabular (16 numeric/cat.) | Term deposit subscription | Marital (Mar/Sin/Div) |
| CelebA | 202,599 | Images (178×218 faces) | Smiling | Gender (M/F) |

Table 1: Summary of datasets used in experiments.

| Metric | Category | Definition |
|---|---|---|
| Accuracy | Utility | $\frac{1}{N} \sum_{i=1}^{N} \mathbf{1}\{\hat{y}_i = y_i\}$ |
| AUROC | Utility | Area under the Receiver Operating Characteristic curve. |
| MI ($u$) | Utility | $\sum_{\hat{y},y} \hat{p}(\hat{y},y) \log \frac{\hat{p}(\hat{y},y)}{\hat{p}(\hat{y})\hat{p}(y)}$    (plug-in estimator) |
| EO_gap | Violation | $\frac{1}{2}\left[(\max_z \mathrm{FPR}_z - \min_z \mathrm{FPR}_z) + (\max_z \mathrm{FNR}_z - \min_z \mathrm{FNR}_z)\right]$ |
| EOpp_gap | Violation | $\max_{z \neq z'} |\mathrm{TPR}_z - \mathrm{TPR}_{z'}|$ |
| CMI ($v$) | Violation | $\sum_y \hat{p}(y) \sum_{\hat{y},z} \hat{p}(\hat{y},z \mid y) \log \frac{\hat{p}(\hat{y},z|y)}{\hat{p}(\hat{y}|y)\hat{p}(z|y)}$ |

Table 2: Evaluation metrics. Empirical probabilities $\hat{p}$ are computed on the held-out test set.

# D  APPENDIX TO SECTION 4

## D.1  DATASETS INFORMATION

See Table 1.

## D.2  METRICS INFORMATION

## D.3  COMPARISON METHODS & IMPLEMENTATIONS

Our baselines are chosen to cover the main existing methods that enforce separation from different approaches: constrained optimization, information-theoretic penalties, distributional robustness, and distribution matching. So that (i) results are (indirectly) comparable to prior or later methods in each of the lines or approaches and (ii) differences among different approaches become visible from the experiments.

For the implementation, we use authors' public code when available; otherwise, we follow their paper pseudocode as closely as possible, keeping model/backbone, splits, and data preprocessing fixed across methods.

**Why these methods.** *EG Reductions* Agarwal et al. (2018) represents the rate-constrained (reductions-based) in-processing family with provable feasibility in expectation. *Adversarial Debiasing* Madras et al. (2018) is the standard minimax approach that enforces conditional independence via an adversary on $Z$. *FR-Train* Roh et al. (2020) and *InfoFair* Kang et al. (2023) are information-theoretic baselines that penalize (conditional) mutual information using variational bounds. *FairDRO* Park et al. (2025) captures distributionally robust optimization for subgroup parity (worst-group risk). *Fair Dummy* Romano et al. (2020) exemplifies distribution matching by constructing a "dummy" sensitive attribute and aligning $(\hat{Y}, A, Y)$ with $(\hat{Y}, \tilde{A}, Y)$. Our method *CMI* directly penalizes an empirical conditional mutual information, providing a single differentiable objective aligned with separation.

Now, we summarize the baseline methods we compare with more detailed explanation:

**Base Empirical-Risk Minimisation (ERM)** minimises the standard cross-entropy with no fairness enforcement. We include the method here to serve as an unconstrained model provides the accuracy ceiling and the EO/EOpp gap floor.

**Threshold Optimiser (Post-processing)** picks group-specific thresholds $\tau_z^{(y)}$ so that $\hat{Y} = \mathbf{1}\{s(x) \geq \tau_Z^{(Y)}\}$ equalizes group error rates. We include this method to represent the line of work that introduces fast "plug-in" repair for any calibrated score $s(x) \in [0, 1]$. Mathematically, it solves a linear program on the empirical distribution to minimize classification cost subject to $\mathrm{TPR}_0 = \mathrm{TPR}_1$ and

| Method | Type | Mechanism |
|--------|------|-----------|
| ERM (Baseline) | – | Unconstrained cross-entropy training. |
| EG Reductions Agarwal et al. (2018) | In | Exponentiated-gradient reduction for rate constraints. |
| Adversarial Debiasing Madras et al. (2018) | In | Min–max game: adversary predicts $Z$ from $\hat{Y}$. |
| FR-Train Roh et al. (2020) | In | Variational lower-bound penalty on $I(\hat{Y}; Z \mid Y)$. |
| InfoFair Kang et al. (2023) | In | InfoNCE bound on latent representation to reduce MI. |
| FairDRO Park et al. (2025) | In | Distributionally robust optimization (worst-group risk). |
| Fair Dummy Romano et al. (2020) | In | Matches distribution to a dummy variable $\tilde{A} \perp Y$. |
| **CMI (Ours)** | In | **Direct, differentiable CMI penalty with gradient norm scaling.** |

Table 3: Comparison methods. "In" denotes in-processing approaches.

$FPR_0 = FPR_1$. With infinite data and calibrated scores, the method attains *exact* EO (Hardt et al., 2016).

**Exponentiated Gradient Reductions (EG)** alternates between (i) training a cost-sensitive classifier $h_t$ and (ii) updating dual weights $w_t$ via multiplicative updates so the weighted errors satisfy EO constraints on expectation:

$$\min_{h \in \mathcal{H}} \mathbb{E}\left[c_w(Y, Z)\mathbf{1}\{h(X) = 1\}\right] \quad \text{s.t.} \quad \mathbb{E}\left[g_j(Y, Z, h(X))\right] \leq 0, \tag{17}$$

optimized by an online convex–concave game with EG updates. After $T$ rounds, $\Delta_{\text{EO}} \leq \mathcal{O}(1/\sqrt{T})$ and risk is within $\mathcal{O}(1/\sqrt{T})$ of the best EO-feasible classifier (Agarwal et al., 2018).

**Adversarial Debiasing** is a popular deep learning baseline as it is easy to add to any network. The method trains a feature extractor $g_\phi$ and label head $h_\theta$ while an adversary $d_\psi$ tries to recover $Z$ from $\hat{Y}$. If $d_\psi$ fails, $\hat{Y} \perp Z \mid Y$.

$$\min_{\phi, \theta} \max_{\psi} \mathcal{L}_{\text{ce}}\left(h_\theta(g_\phi)\right) - \lambda \mathcal{L}_{\text{adv}}\left(d_\psi(h_\theta(g_\phi)), Z\right). \tag{18}$$

If the inner adversary is optimal and the game converges, $I(\hat{Y}; Z \mid Y) = 0 \Rightarrow$ EO holds (Madras et al., 2018), but no finite-sample bound.

**FR-Train (Mutual-Information Regularizer)** directly penalizes $I(\hat{Y}; Z \mid Y)$:

$$\min_{\theta} \mathcal{L}_{\text{ce}}(\theta) + \lambda \widehat{I}_{\text{NWJ}}(\hat{Y}_\theta; Z \mid Y). \tag{19}$$

A minibatch estimator of CMI is added to the loss so that gradients push predictions toward independence. With a consistent MI estimator and $\lambda \to \infty$ the optimizer converges to EO; generalization bounds of order $\mathcal{O}(\sqrt{\frac{\log n}{n}})$ on the EO gap were derived by Roh et al. (2020).

**FairDRO** is a distributionally-robust method that improves worst-group error. It minimise risk under the *worst-case* class-conditional re-weighting, forcing the model to raise minority-group TPR:

$$\min_{\theta} \max_{w \in \mathcal{W}} \sum_i w_i \ell\left(f_\theta(x_i), y_i\right), \quad \text{s.t.} \sum_i w_i = 1, \ w_i \leq \frac{\rho}{n_y} \tag{20}$$

where $\rho$ controls robustness radius. Optimal classifier matches the EO-relaxed benchmark $\min_\theta \max_z \text{Err}(z)$, with convergence rate $\mathcal{O}(1/\sqrt{n})$ (Park et al., 2025).

**Fair Dummy (Equi-Class Learner)** forces its predictions to be indistinguishable across $Z$ by a discriminator operating on *feature* representations rather than logits. A light-weight two-network game that equalises class-conditional means: Alternating optimisation of classifier $f_\theta$ and discriminator $d_\psi$ with objective $\mathcal{L}_{\text{ce}}(\theta) - \lambda \text{MMD}\left(f_\theta(X), Z\right)$. Converges to an EO-feasible solution when the discriminator has infinite capacity; no finite-sample rate is known.

**CMI-NN (Ours)** Plug-and-play regulariser that retains deep-net accuracy. Adds an information-theoretic penalty directly on logits, trained end-to-end with Adam:

$$\min_{\theta} (1 - \lambda)\mathcal{L}_{\text{ce}}(\theta) + \lambda \widehat{I}(\hat{Y}_\theta; Z \mid Y), \qquad \lambda \in [0, 1]. \tag{21}$$

We prove that for any $\delta > 0$, $I(\hat{Y}; Z \mid Y) \leq \delta \implies \Delta_{\text{EO}} \leq \sqrt{2\delta}$. Hence EO gap shrinks at the *same* rate as the MI penalty.

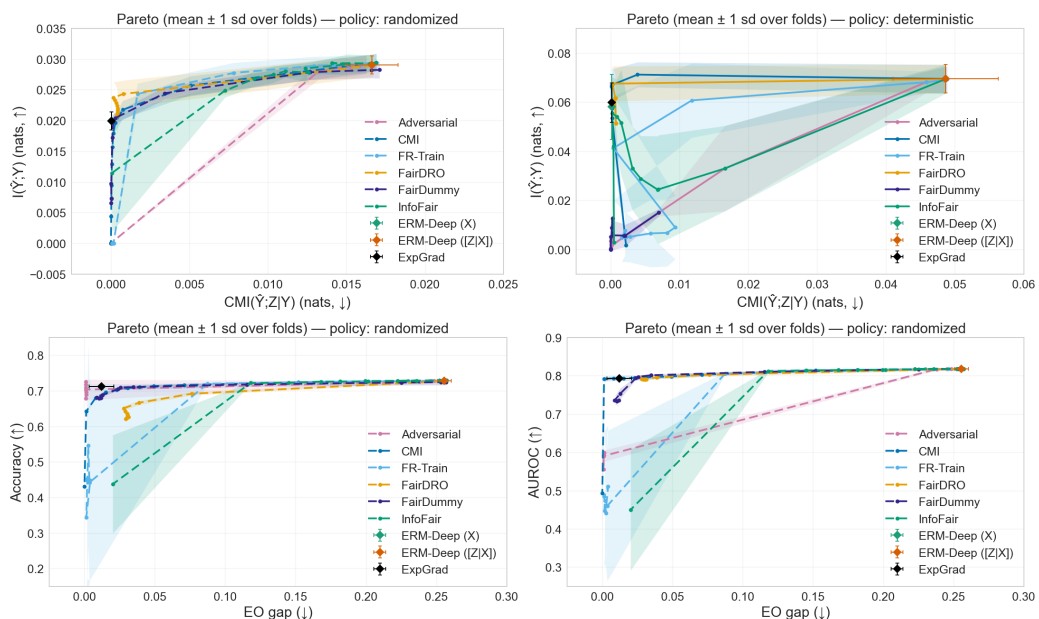

Figure 3: **Adult Results: Low Marginal Cost and Robust Transfer. Top (Information Plane):** In contrast to Bank, the Adult frontier exhibits a *low marginal cost of separation*; high utility is preserved even as the violation approaches zero ($v \approx 0$). Normalized CMI traces a smooth, concave envelope, avoiding the non-monotonicity and variance observed in proxy-based baselines. **Bottom (Operational Transfer):** This favorable geometry transfers to deployment metrics. CMI achieves near-optimal Accuracy and AUROC even at negligible EO gaps, confirming that when the theoretical cost of fairness is low, the direct estimator can reliably recover the optimal trade-off.

## D.4 EXPERIMENT RESULTS AND ANALYSIS ON ADULT DATASET

Figures 3 present the results on the Adult dataset.

- **Concave Frontier Verification:** The utility-violation curves in Fig. 3 top are approximately concave, corroborating Theorem 2.2. Notably, the *Randomized* frontiers (representing the convex hull of policies) encompass the *Deterministic* frontiers, confirming that randomization is required to reach the theoretical optimum. CMI traces the outermost (dominant) frontier, indicating it learns a superior representation compared to baselines.
- **Estimation Stability:** A key contribution of our gradient-normalized algorithm is stability. CMI produces a smooth, monotonic frontier with minimal variance (tight error bands) across folds. In contrast, *FairDRO* and *Adversarial Debiasing* exhibit kinks and non-monotonic behavior, where increasing the fairness weight occasionally degrades utility without improving fairness, or vice versa.
- **Generalization:** Although CMI optimizes the information-theoretic bound, Fig. 3 bottom shows this transfers perfectly to operational metrics. CMI achieves the highest AUROC for any given EO gap, effectively dominating the trade-off landscape.

## D.5 EXPERIMENT RESULTS AND ANALYSIS ON COMPAS DATASET

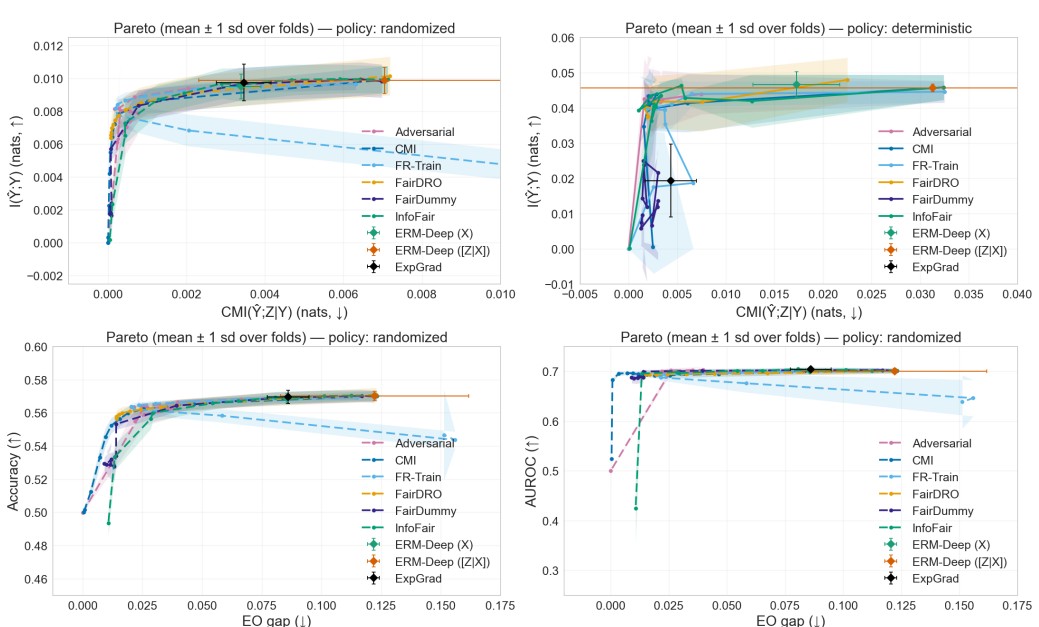

Figure 4: **COMPAS Results: Strict-Regime Coverage and Stability. Top (Information Plane):** Normalized CMI generates a coherent, well-ordered Pareto traversal into the strict regime ($v \approx 0$), avoiding the fold-level variance and non-Pareto artifacts (e.g., non-monotonic segments) observed in several proxy-based baselines on this noisy benchmark. **Bottom (Operational Transfer):** The stability advantages are most pronounced in the strict-fairness region. As the EO gap is tightened, CMI retains high AUROC and Accuracy, whereas baselines suffer sharper degradation or require larger operational gaps to recover comparable utility.

