# OpenReview forum: "Separation-Utility Pareto Frontier: An Information-Theoretic Characterization"
_ICLR.cc/2026/Workshop/AFAA — Submitted to AFAA 2026_

### Official Review · Reviewer_LZpm · 2026-02-19
**Limited Novelty**

**Rating:** 1
**Confidence:** 5

**Summary:**

The goal of this paper is to study the trade-off between utility and the fairness metric separation (independence from sensitive attributes conditioned on the true outcome). They leverage the information-theoretic measure conditional mutual information $I(\hat{Y}; Z|Y)$ to quantify separation and incorporate it as a regularizer.

**Strengths:**

Several datasets have been considered.

**Weaknesses:**

The idea of quantifying separation using CMI or incorporating CMI as a regularizer is not fundamentally new in algorithmic fairness and has been widely studied in several prior works. Many closely related works have been missed. I would recommend including these references and highlighting the novelty in comparison to these prior works.

For instance, the core idea of representing separation $\hat{Y} \perp Z| Y$ as $I(\hat{Y}; Z|Y)$ has appeared in the following published prior works (Also see the references therein):

[1] Fairness in supervised learning: An information theoretic approach. https://arxiv.org/abs/1801.04378 (goes back to ISIT 2018)

[2] Fairness Under Feature Exemptions: Counterfactual and Observational Measures (AAAI) https://arxiv.org/pdf/2006.07986

In fact, [2] already considers the CMI term as a regularizer, and should be included as a baseline. Also see the references therein.

Other recent related works have also used information-theoretic characterizations of all three primary fairness measures - independence (MI), separation (CMI), and sufficiency (CMI), and considered their tradeoffs often with utility. Compared to these, the novelty of the current paper studying the tradeoff between only one fairness measure separation and utility using information theory seems to be quite limited in scope. (Also see the references therein).

[3] Gradual (In)Compatibility of Fairness Criteria (AAAI) https://arxiv.org/abs/2109.04399

Another work uses Pinsker's inequality to prove a relation very similar to Theorem 2.5 in the current paper.

[4] Demystifying Local and Global Fairness Trade-offs in Federated Learning Using Partial Information Decomposition https://arxiv.org/pdf/2307.11333

The use of CMI as a regularizer has also been studied in the following paper and several other subsequent works that reference this work. The CMI considered is $I(\hat{Y};Z|X')$ where $X'$ is some feature, but it can very well be $Y$, and the technique would not change. I would encourage comparison with these baselines, including [2] and [5], and several other subsequent works that reference them.

[5] Algorithmic decision making with conditional fairness https://arxiv.org/abs/2006.10483

These are just some of the main references, but there are several other works that consider CMI (also called conditional statistical parity) as a regularizer, which would technically be the same if $X'=Y$. I would encourage discussing all these related works, and highlight the technical difference of the way CMI is being incorporated in the current work as compared to these prior works.

Other relevant papers on fairness utility tradeoffs using information theory:

[6] Inherent Tradeoffs in Learning Fair Representations https://arxiv.org/abs/1906.08386

[7] Aleatoric and Epistemic Discrimination: Fundamental Limits of Fairness Interventions

---

### Official Review · Reviewer_j6UG · 2026-02-20
**A Conditional Mutual Information (CMI) based Framework for Utility-Separation (Fairness) Trade-offs**

**Rating:** 4
**Confidence:** 3

**Summary:**

This paper studies the trade-off between utility and separation -fairness- (equalized odds) in binary classification through an information-theoretic perspective. Separation violation is quantified using Conditional Mutual Information (CMI), while utility is measured via mutual information with the target label. The authors characterize the achievable utility-separation region and show that allowing randomized predictors yields the concave closure of the deterministic Pareto frontier, explaining its geometry and marginal cost behavior. Based on this analysis, they propose a direct CMI-based regularization approach for training fair models and empirically demonstrate stable and competitive trade-offs on well-known fairness benchmark datasets (Adult, Compas, Bank, CelebA).

**Strengths:**

The strengths of the paper are listed as below:

1. The paperwork provides a clean theoretical characterization of the utility-separation Pareto frontier and clarifies the role of randomization.
2. The study offers a principled justification for CMI as a scalar measure of separation violation with an auditing interpretation.
3. The framework proposes a simple and stable in-processing regularizer aligned with the theoretical analysis.
4. Empirical results show smooth and stable Pareto frontiers across multiple datasets and evaluation settings.
5. The code will be released upon acceptance so the reproducibility is possible.

**Weaknesses:**

The weaknesses of the paper are listed as below:

1. The plug-in CMI estimator is assumed to be reliable for discrete tasks; clearer discussion of its limitations (sparsity, continuous scores etc.) would improve practical guidance.
2. Some theoretical assumptions underlying strict trade-offs may not hold in real datasets and could be better contextualized.
3. Evaluation relies mainly on visual Pareto curves qualitatively; there are still needs to have quantitative analysis to compare different trade-off methods. Hypervolume (https://pymoo.org), Fairical (https://pypi.org/project/fairical) or some other tools may be used to quantify and compare pareto frontiers of the models.
4. Using MLP-based backbone for each model in comparison may give a limited understanding of the generalizability of the proposed approach.

---

### Official Review · Reviewer_AULX · 2026-02-21
**Review for submission 28**

**Rating:** 4
**Confidence:** 4

**Summary:**

This work connects theoretical fairness constraints with practical optimization through a model-agnostic description of the separation–utility Pareto frontier. It proves that the frontier is concave and shows that, for classification, complex learned proxies are often not needed. Instead, a direct empirical estimator of Conditional Mutual Information produces smoother frontiers, lower variance, and inherently fairer representations than more sophisticated baseline methods.

**Strengths:**

The paper is very well-written and easy to follow; the theoretical results are rigorous, and the intuitions are clearly explained.

The paper rigously studies the separation–utility Pareto frontier behind just the binary setting, which is important and is currently missing in the fairness literature.

**Weaknesses:**

Minor:
1) line 89: citation on Sagawa et al has an extra * in it.
2) CMI is used during section 1 but is only properly defined later in section 2.


Question:
1) Why are some lines in the figures zigzagging?

---

### Meta-Review · Area_Chair_sovy · 2026-02-26

**Recommendation:** Reject
**Confidence:** 3

**Metareview:**

This paper provides an information-theoretic characterization of the utility-separation Pareto frontier, demonstrates its concavity, and proposes a direct CMI-based regularizer for fair deep learning. While the theoretical results are well explained and the empirical evaluation covers several standard datasets, the work's novelty is significantly limited. As noted by Reviewer LZpm, the core concepts of using CMI as a fairness measure and regularizer have been extensively explored in prior literature, and this submission lacks critical comparisons with these established similar proposals.

Furthermore, the exploration of the "Pareto Frontier" mentioned in the title is insufficient for a main track submission. While visual frontiers are presented, the paper fails to utilize standard Multi-Objective Optimization (MOO) metrics—such as hypervolume or specific Pareto-based comparisons—to quantitatively assess the trade-off landscapes against baselines.

---

### Decision · Program_Chairs · 2026-03-02

Reject